# ACPBench Hard: Unrestrained Reasoning about Action, Change, and Planning

**Harsha Kokel,**
IBM Research
San Jose, California 95141, USA
`Harsha.Kokel@ibm.com`

**Michael Katz, Kavitha Srinivas & Shirin Sohrabi**
IBM Thomas J. Watson Research Center
Yorktown Heights, New York 10598, USA
`{Michael.Katz1,Kavitha.Srinivas}@ibm.com,`
`ssohrab@us.ibm.com`

## Abstract

We introduce ACPBench Hard, a dataset of generative, open-ended questions which LLM models needs to answer in order to plan. Models that perform well on these tasks could in principle be integrated into a planner or be used directly as a policy. We discuss the complexity of these tasks as well as the complexity of validating the correctness of their answers and present validation algorithms for each task. Equipped with these validators, we test the performance of a variety of models on our tasks and find that for most of these tasks, the performance of even the largest models is still subpar. The models do not possess even the most basic capability of identifying which actions can be performed in a given state. No model outperforms any other on our proposed tasks and, with a few exceptions, all tested language models score below 65%, indicating that even the current frontier language models as well as so-called reasoning models have a long way to go before they can reliably reason about planning. [1].

## 1 Introduction

The ability to reason and plan in large language models (LLMs), is a major focus in the field. For reasoning, the majority of work focuses on the mathematical reasoning (Cobbe et al., 2021) and logical inference (Saparov & He, 2023). For planning, most work focused on the ability to produce or validate a plan (Valmeekam et al., 2023a; Stein et al., 2026). The downside of focusing on the end-to-end planning datasets is the inability to pinpoint the reason for a black-box planner, such as an LLM-based one, to not be able to produce a solution.

To tackle this gap, ACPBench (Kokel et al., 2025) introduced a benchmark for testing the reasoning abilities about action, change, and planning, separating the planning process into the atomic reasoning tasks performed by planners. Tasks in these benchmarks are boolean or have multiple choice. However, in real applications planners do not typically have shortlisted options. Planners need to generate the actions from significantly larger action space. For instance, the applicability task in ACPBench either asks about applicability of a given action in the boolean case or chooses an applicable action among 4 variants. Even if we make the assumption that the model can answer such questions with high precision, it is not clear how this capability can be efficiently exploited for the task of generating all applicable actions in a given state.

In this work, we create a generative version of ACPBench to alleviate these limitations. We devise open-ended, generative versions of questions for the same 7 tasks, and add a new challenging task of finding an action that takes us closer to the goal, a task that corresponds to the ability to perform *optimal* planning (Bylander, 1994). For each of these tasks, we introduce an evaluator, which scores a possible answer to the open-ended questions. We generate an evaluation set of questions based on a wide collection of 13 Planning Domain Definition Language (PDDL) (McDermott, 2000) domains from ACPBench, which we call *ACPBench Hard*. The core idea is to evaluate the LLMs' ability to produce reliable components for automated planners, as LLMs that perform well on these tasks could be integrated into a planner or be used directly as a policy (Kambhampati et al., 2024).

---

[1]ACPBench Hard collection is publicly available, see `https://ibm.github.io/ACPBench`

For all tasks, we discuss their computational complexity, as well as the complexity of validating their solutions. We devise validators for each task, based on the symbolic description of the questions. With the help of these validators, we test the performance of a collection of modern LLMs of various sizes on ACPBench Hard. We find that the performance of language models, even the largest ones, is still insufficient to be reliably used in planners, see Figure 1. We observe that there is no single model that outperforms all other models on all tasks of the ACPBench Hard dataset. Further, on half of the tasks, namely "reachability" (reach), "action reachability" (areach), "landmarks" (land), and "applicability" (app), all tested language models exhibit a very low accuracy. All tested models score below 65% on most tasks, indicating that even the current frontier language models have a long way to go before they can reliably reason about planning. Further, even the so-called reasoning models o1 perform poorly on half of these tasks. The o1-preview model achieves 89% on the "progression" (prog) task and 80% on the "next action" (nexta) task, with the rest of the results being 66% and below, and the smaller o1-mini model outperforms it only on the "plan validation" (val) task with 78% accuracy. The much higher computational effort of the reasoning models compared to the language models do not seem to justify the somewhat moderate increase in accuracy.

## 2 BACKGROUND

We consider planning tasks $\Pi = \langle F, A, s_0, s_\star \rangle$ in the STRIPS formalism (Bylander, 1994). In such a task, $F$ is a set of Boolean *propositions*. Each subset $s \subseteq F$ is called a *state*, and $S = 2^F$ is the *state space* of $\Pi$. The state $s_0$ is the *initial state* of $\Pi$. The goal $s_\star \subseteq F$ is a set of propositions, where a state $s$ is a *goal state* if $s_\star \subseteq s$. The set $A$ is a finite set of *actions*. Each action $a \in A$ has an associated set of *preconditions* $pre(a) \subseteq F$, *add* effects $add(a) \subseteq F$ and *delete* effects $del(a) \subseteq F$.

The semantics of STRIPS planning is as follows. An action $a$ is *applicable* in the state $s$ if $pre(a) \subseteq s$. Applying $a$ in $s$ results in the state $s[\![a]\!] := (s \setminus del(a)) \cup add(a)$. A sequence of actions $\pi = \langle a_1, \ldots, a_n \rangle$ is *applicable* in $s$ if there exists a sequence of states $s = s_1, \ldots, s_{n+1}$ such that for each $1 \leq i \leq n$ we have $a_i$ is applicable in the state $s_i$ and applying it results in the state $s_{i+1}$. If it exists, such sequence is uniquely defined, and its end state $s_n$ is denoted by $s[\![\pi]\!]$. An applicable action sequence is a *plan* for $s$ if $s[\![\pi]\!]$ is a goal state. A plan for $s$ with minimal length is called *optimal*. The *perfect heuristic* for $s$, denoted by $h^*(s)$, is the cost of an optimal plan for $s$. The objective of (optimal) planning is to find an (optimal) plan for the state $s_0$.

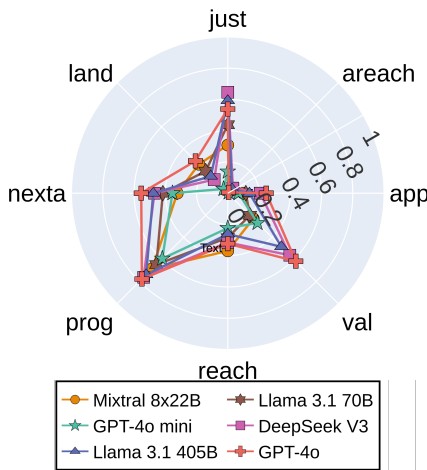

Figure 1: Task-wise accuracy of large size language models.

## 3 RELATED WORK

The most relevant to our work is the ACPBench dataset (Kokel et al., 2025), which we build upon. It features 7 core reasoning tasks about planning. ACPBench does not capture the decisions that the automated planners need to make, as these decisions are generative in their nature. In this work, therefore, we present the generative form of these tasks, requiring LLM to produce precisely the answers that automated planners produce and consume. Additionally, we present a new task, not considered by previous work, requiring to produce an action that takes us closer to the goal.

Other notable work in similar direction includes Textual Reasoning about Actions and Change (TRAC) (He et al., 2023), featuring 4 tasks: projection, execution, planning, and goal recognition, as well as PlanBench (Valmeekam et al., 2023b), with 8 planning tasks including plan generation, reasoning about plan execution, and plan verification. Both benchmarks focus on a small number of planning domains (mostly BlocksWorld and variants). Both use templates to generate natural language text; similar to ours. AutoPlanBench (Stein et al., 2025) alleviates the dependence on templates, leveraging LLMs to generate these natural language template and hence were able to scale up the dataset to 12 domains. While scaling the number of domains, they limit their focus on planning task including plan and next-action generation.

Another notable dataset is ActionReasoningBench (Handa et al., 2025), featuring six tasks: fluent tracking, state tracking, action executability, effects of actions, numerical RAC, and composite questions. These tasks reason about action sequences and hence two of the tasks overlap with three of the tasks we consider. Specifically, action executability deals with questions that fall under applicability and validation tasks in our case. On one hand, this creates a complicated reasoning question, on the other, it makes it harder to pinpoint the source of an error. The effects-of-actions task is somewhat similar to our progression task, applied repetitively. Further, ActionReasoningBench uses a large-language model for evaluation of the generative questions, which is not quite robust. We develop dedicated symbolic validators to evaluate each of the generative task introduced. Table 4 in the Appendix compares and contrasts ACPBench Hard with related planning benchmarks. Research on planning abilities of agents (Liu et al., 2024; Ma et al., 2024) is also relevant. In our experiments, we focus on the planning abilities of individual models instead of agents. However, the "next action" prediction task does takes us in direction of agents.

## 4 Underlying Reasoning Tasks

We focus on the core reasoning capabilities that symbolic planners routinely deploy when deciding how actions interact with states and how plans progress toward goals. These capabilities span action-level, state-level and plan-level reasoning. The tasks range from identifying what actions are possible right now, to anticipating future possibilities, to validating multi-step plans. Each task probes a different facet of an agent's ability to reason about actions, state transitions, and goal achievement.

At the action level, **applicability** tests whether a model identifies all actions that are valid and applicable in current state. Failure in this reasoning often surface as LLMs hallucinating non-existent tools/parameters (Lin et al., 2025), pick objects that are not present, move through blocked spaces (Raman et al., 2024).

**Progression** checks all the state updates are predicted for an applicable action; typical errors in this reasoning include missing/deleting the wrong facts, tracking failures like forgetting IDs, assuming deleted entities persist, or ignoring side effects (c.f. tracking objects task in Suzgun et al. (2023)).

At state-level, **Reachability** evaluates the ability to determine which facts could eventually become true and which facts can never become true along any valid trajectory from the current state; failures arise when models attempt to achieve structurally impossible facts or invent intermediate states.

**Action reachability** evaluates the ability to determine which actions could eventually become applicable and which actions can never become applicable in any state reachable from the current state; failures occur when models propose actions whose preconditions can never co-occur, leading them to explore permanently unreachable or irrelevant branches, or supplying invalid arguments to actions that make it inapplicable in any reachable state (Xu et al., 2023).

At the plan level, **validation** checks whether a plan is executable and identifies the earliest inapplicable action; failure in this reasoning leads to the model missing unmet preconditions and erroneously deeming infeasible steps as valid.

**Justification** tests whether a plan can be simplified by removing redundant actions; failure in this reasoning leads to an unnecessarily elaborate plan with redundant steps or an overly compressed plan that removes necessary actions and breaks plan validity.

**Landmarks** require identifying facts that must inevitably hold along every valid solution path; failure in this task can manifest as model skipping mandatory subgoals or collapsing multi-step dependencies required for achieving the goal.

Finally, **next-action selection** evaluates whether the model can chose an action that meaningfully reduces goal distance; errors include drifting toward irrelevant or harmful actions, or selecting steps that do not improve the optimal cost.

## 5 Dataset Construction

Building upon ACPBench (Kokel et al., 2025), we keep the same 13 planning domains and create generative dataset. For each task, we describe the data stored in order to enable or speed up the

```
Context: There are several cities, each containing several locations, some of which are airports. There
  are also trucks, which can drive within a single city, and airplanes, which can fly between airports.
  The goal is to get some packages from various locations to various new locations. There are 2 trucks
  and 1 airplane, as well as 4 packages. There are 4 locations across 2 cities. The locations are in
  cities as follows: l1-1 and l1-0 are in c1; l0-1 and l0-0 are in c0. Currently, p2, t1, p1, p3, a0, and
  p0 are at l1-0, t0 is at l0-1. The available propositions are:  (at ?obj ?loc) - ?obj is at ?loc and
  (in ?obj1 ?obj2) - ?obj1 is in ?obj2.

Inputs: Break down the outcomes of performing the action "load object p3 into truck t1 at location l1-0"
  into two lists, positive effects and negative effects. Positive effects are the propositions that are
  false in the current state but will become true after performing the action. Negative effects are the
  propositions that are true in the current state and will become false after performing the action.
```

Figure 2: Example of a question for the progression task in ACPBench Hard. (Also see examples in Appendix.)

evaluation of the correctness of a potential answer. Often, the PDDL planning task $\Pi = \langle F, A, s, s_\star \rangle$ is needed for evaluation, and therefore we store it per question, with the current state $s$ as the initial state. Plans, whenever needed, are computed with a classical planner. We first attempt to run a top-quality planner (Katz & Lee, 2023b), producing plans of best possible quality, and use a diverse planner (Katz & Sohrabi, 2020) as a backup mechanism if no top quality plans were found. The high-level construction workflow is illustrated in the Figure 7. We use template-based approach to convert the PDDL to NL.

**1. Applicability (App)** The first task deals with identifying which actions are applicable in a state. For an action to be applicable, its preconditions must hold in the state. Given a state $s$ and the set of actions $A$, the subset of applicable actions would be $A(s) = \{a \in A \mid pre(a) \subseteq s\}$, easily computable by iterating over the actions. The complexity of such an iterative algorithm is $O(|F||A|)$, since $|F|$ is a theoretical upper bound on the precondition size. In practice, planning problems typically have small preconditions size. Thus, we can create a generative question by simply asking the model to produce all applicable actions in a given state. For practical reasons, we impose a bound on the number of applicable actions in a state, generating questions only in cases when $|A(s)|$ is under that bound. We can keep all applicable actions names in the dataset for validating the answer. The number of applicable actions for all instances is $< 10$, except for Swap and Alfword domains (ref. Table 5 in the Appendix).

**2. Progression (Prog)** The next task evaluates LLMs ability to understand how the world state changes by action application. Performing an action changes the state in the following manner: The delete effects will no longer hold and the add effects will hold. Everything else remains unchanged. Given a state $s$ and an action $a$, the next state is $t = (s \setminus del(a)) \cup add(a)$. The complexity of the straightforward computation is $O(|F|)$ worst case, but in practice the add and delete effects of planning problems are typically small. We construct a single generative question, asking what propositions are false in the current state but will become true after performing the action and asking which ones are true and become false. The first set is $t \setminus s$, and the second one is $s \setminus t$. We maintain both of these sets in the answers. Figure 2 shows an example of a progression question.

**3. Reachability (Reach)** The reachability task evaluates if a specific fact can eventually become true by taking (possibly multiple consecutive) actions in the given state. This is a multi-step reasoning task that can help avoid exploring unfeasible options. The generative version of this question is quite simple: what proposition can never hold in any potentially reachable state. If no such propositions exist, we instruct to reply *None*. Reachability is PSPACE-hard to answer in general (Bylander, 1994) for partial states, so we only focus on reachability of a specific fact. We can generate some unreachable facts by either finding groundings of *static* predicates (unchanged by any action) that do not hold in the given state, or by (under)approximating the reachability with poly-time computable delete-relaxed reachability (Hoffmann & Nebel, 2001). For practical reasons, we keep a subset of generated unreachable facts, $N$, for speeding up answer validation. We generate question only in states where we either *know at least one unreachable fact* or we *know that all facts are reachable*. In case of doubt, we skip generating questions for that state.

**4. Action Reachability (AReach)** The action reachability task is closely related to the (atom) reachability, checking whether there is a reachable state where the action is applicable. Computationally, this problem is PSPACE-hard, for the same reason as (atom) reachability. The generative version of the question is: what action can never become applicable, in any state reachable from the current state? Here as well, we generate question only in states where we either *know at least one unreachable action* or we *know that all actions are reachable*. In case of doubt, we skip the state. For answer

validation, as in the previous case, we keep a set of example unreachable actions $U$. Set $U$ is empty when all actions are reachable.

**5. Validation (Val)** The validation task aims at checking whether the specified sequence of actions $\pi$ is a plan. In other words, whether $\pi$ is valid, applicable, and successfully achieves the intended goal from the given state $s$. The generative version of this question aims at identifying where the plan fails. In other words, we ask to identify the first inapplicable action in a given sequence of actions. To generate such a sequence, we start with a valid plan and randomly choose an action to replace with an inapplicable one. For validation purposes, we keep the index of such action.

**6. Justification (Just)** Action justification (Fink & Yang, 1992; Salerno et al., 2023) deals with the question of whether a given plan can be simplified by removing some actions. The generative version of the justification task question asks to simplify the plan by removing one or two consecutive actions and to produce the resulting simplified plan. To produce such a sequence, we start with a plan for the initial state. We check whether the plan can be simplified by removing a single or two consecutive actions. If not, we try to extend the plan by adding such an action or a pair of actions, keeping the resulting sequence a plan. For answer validation, we keep the action or pair of actions that can be removed, together with the appearance number, for actions that appear more than once on the plan.

**7. Landmarks (Land)** Landmarks task tests LLM's ability to identify subgoals that are necessary to achieve the goal. In the planning literature such subgoals are often called landmarks (Porteous et al., 2001). Landmarks are facts that must become true sometime along every plan. While checking whether a proposition is a landmark is PSPACE-hard (Porteous et al., 2001), there are several methods that can find a subset of landmarks (Keyder et al., 2010; Hoffmann et al., 2004; Richter et al., 2008; Zhu & Givan, 2003). We use the so-called RHW method (Richter et al., 2008). Further, negative evidence can be obtained from a collection of plans - a proposition that does not appear on a plan is not a landmark. We keep the sets of facts that are known to be landmarks and of facts that are known to be non-landmarks for speeding up answer validation.

**8. Next Action (NextA)** An additional task that does not appear in ACPBench is the next action task. This generative question asks what is the next action that takes us towards the goal. This task is closely related to optimal planning, since optimal plans can be produced by iteratively obtaining such actions. While even non-optimal planning is PSPACE-hard (Bylander, 1994), modern planners can often quickly find collections of optimal plans (Katz et al., 2020; Katz & Lee, 2023a). Clearly, the first actions of these plans would be correct answers. Further, the cost of these optimal plans can be used to check other applicable actions in the state, by producing an optimal plan for the states obtained by applying these actions in the question state. We keep the sets of actions that are correct answers and of actions that are known to not take us closer to the goal for speeding up answer validation. Further, we store the optimal cost $h^*(s)$ of achieving the goal from the current state.

## 6 ANSWER EVALUATION

Evaluating answers to boolean or multiple-choice questions simply amounts to looking up the correct answer and comparing. Evaluating open-ended answers, on the other hand, might not be as easy. This is due to the fact that sometimes, there is not one single correct answer that can be stored with the question. Looking at the tasks at hand, while in some cases we can store the complete correct answer, in other we must resort to performing some computation in order to evaluate whether the returned answer is correct. In what follows, we describe how the answer is evaluated for each task.

**1. Applicability (App)** In this task, we store all applicable action names per question. The answer evaluation therefore amounts to a simple comparison between the given answer and the correct one. Since we ask for all applicable actions, we chose to assign a score 1 if the set of all actions in the answer equals to the set of all applicable actions. Otherwise, we assign 0. The complexity of validating the answer is therefore $O(1)$ if we impose a constant threshold on the number of applicable actions when the question is created, otherwise it is $O(|A|)$.

**2. Progression (Prog)** Here, we store both correct sets of propositions ($t \setminus s$, and $s \setminus t$) per question. An answer validation amounts to a simple comparison between the given answer and the correct one. Since we test the ability to produce all action effects, we score 1 if both all positive and all negative effects were correctly identified. Otherwise, we give the score 0. The complexity of validating the answer is therefore $O(|F|)$.

**3. Reachability (Reach)** The set of unreachable facts $N$ being empty is an indication that there are no unreachable facts. Hence, if the answer $P$ is *None*, we assign score of $1$ when $N = \emptyset$; otherwise, the score is $0$. If the answer $P$ is not *None* and $P \in N$, we assign the score of $1$. If $P$ is not *None* but the set $N$ is empty, the score is $0$. In the remaining case, where $P$ is not part of $N$, we need to solve a planning task $\Pi' = \langle F, A, s_0, \{P\} \rangle$, where the goal is to achieve the atom $P$ and the rest as in the planning task in the question. We can use any off-the-shelf planner to generate a plan for this task. If a plan exists, we score the answer $0$, otherwise $1$. Therefore, the evaluation problem in this case is PSPACE-complete.

**4. Action Reachability (AReach)** As in the previous case, when the set of unreachable actions $U$ is empty, all actions are reachable. Hence, if the answer is *None*, we assign a score of $1$ when $U = \emptyset$; otherwise the score is $0$. For an action $a$ as answer, if $U = \emptyset$ then the score is $0$. If $a \in U$, the score is $1$. In the remaining case, where $U \neq \emptyset$ and $a \notin U$, we construct a planning task $\Pi' = \langle F, A, s_0, pre(a) \rangle$ with the goal being the preconditions of the action $a$, and the rest as in the planning tasks in the question. We run a planner on this task, checking if a plan exists. If yes, we score the answer $0$, otherwise $1$. The evaluation is, therefore, PSPACE-complete in this case as well.

**5. Validation (Val)** In this case, we only need to compare the index in the answer to the correct index of the first inapplicable action. We give a $0/1$ score based on that comparison. The complexity of validating the answer is therefore $O(1)$.

**6. Justification (Just)** To check whether the returned sequence is correct, we slightly relaxed the constraint in the question. We check whether it is a proper subsequence of the provided plan and whether it is a plan. If both are true, we give a score of $1$, otherwise, we score the answer $0$. The complexity of validating the answer is therefore $O(|\pi||F|)$ for the plan $\pi$.

**7. Landmarks (Land)** Invalid propositions are scored $0$. A proposition $p \in s_0 \cup s_\star$ is a trivial landmark and is also scored $0$. Given a valid proposition $p \in F \setminus (s_0 \cup s_\star)$, if it is known not to be a landmark (e.g., there exists a plan that does not traverse any state in which $p$ holds), we assign it a score of $0$. Otherwise, in order to check if $p$ is a non-trivial landmark, we construct a planning task $\Pi' = \langle F', A', s_0', s_\star' \rangle$ as follows. The set of propositions is extended with a proposition $p_{ach}$ that is intended to indicate that $p$ was never achieved along a sequence of actions. $F' = F \cup \{p_{nach}\}$ Thus, for each action $a \in A$ such that $p \in add(a)$, we construct a new action with extended delete effect to include $p_{nach}$. Formally, for an action $a$, $a'$ is defined as follows. $pre(a') = pre(a)$, $add(a') = add(a)$, and $del(a') = del(a) \cup \{p_{nach}\}$ if $p \in add(a)$, otherwise $del(a') = del(a)$. The extended action set is therefore $A' = \{a'\} \mid a \in A$. Finally, both the initial state and the goal are extended with $p_{nach}$: $s_0' = s_0 \cup \{p_{nach}\}$, $s_\star' = s_\star \cup \{p_{nach}\}$, indicating that we are interested in plans that do not achieve $p$ along their path, as any action that achieves $p$ will delete $p_{nach}$ from the state, making the goal not reachable. We use an off-the-shelf planner to check if there is a plan for $\Pi'$. If there is one, it corresponds to a plan for $\Pi$ that does not make $p$ true and therefore $p$ is not a landmark, and we assign the score of $0$. If $\Pi'$ is found unsolvable, then $p$ is a landmark and we assign the score of $1$. The answer validation is therefore PSPACE-complete in this case as well.

**8. Next Action (NextA)** For an action $a$, if it is in the set of kept correct answers, we score it $1$ and if it is in the set of known incorrect answers, we score it $0$. Otherwise, if it is applicable, we apply it to the current state $s$ and obtain the state $t$. We find the optimal plan costs $h^*(s)$ and $h^*(t)$ for these two states with the help of an optimal planner and score the answer $1$ if $h^*(s) - h^*(t) = 1$. Otherwise, we score it $0$. The answer validation is therefore PSPACE-complete in this case as well.

## 7 EXPERIMENTS

### 7.1 BENCHMARKING PERFORMANCE

Following ACPBench, we generated 10 open-ended questions per domain for each of the 8 tasks, a total of 1040 questions. We evaluated 15 language/reasoning models, with the aim to cover small, medium, and large models, *Small size*: Granite 3.1 8B (Team, 2024) and Llama 3.1 8B (Dubey et al., 2024); *medium size*: DeepSeek coder 33B (Guo et al., 2024) and Granite 34B code (Mishra et al., 2024); *Large size*: Mixtral 8x22B (MistralAI, 2024), Llama 3.1 70B and Llama 3.1 405B (Dubey et al., 2024), DeepSeek V3 (DeepSeek-AI et al., 2025b), GPT-4o mini, GPT-4o (OpenAI et al., 2024a); *Reasoning* DeepSeek R1 (DeepSeek-AI et al., 2025a), o1 mini, o1-preview

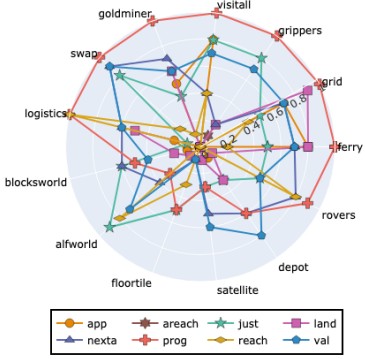

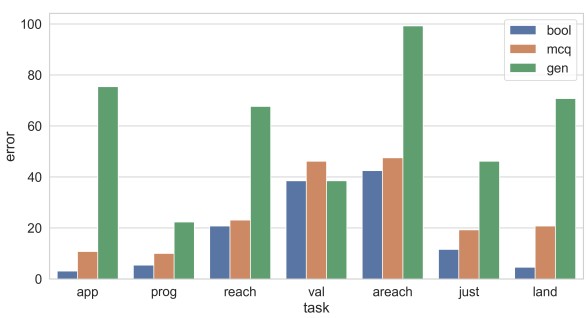

Figure 3: Domain-wise accuracy of GPT-4o.

Figure 4: Comparison of prediction error for GPT-4o on different question formats.

(OpenAI et al., 2024b), and recent open-weight reasoning models from OpenAI—GPT OSS 20B and 120B (OpenAI et al., 2025).

All models were either accessed using API or hosted locally using hugging face transformer library on machines with 2 $A$100 80 GB GPU. It is important to note that there is a significant difference in the energy consumption and evaluation cost between various models. While smaller models are relatively cheap, the larger language models such as Llama 3.1 405B and GPT-4o are prohibitively expensive to be used as planner components. The reasoning models such as o1 and even the cheaper DeepSeeek R1 are even more expensive than the language models. Therefore, our experiments with these reasoning models are intended mostly for providing a frame of reference.

For each task, we evaluated a 2-shot prompting with static examples from outside the evaluation set. The two examples are from the grid and logistics domains, one each per task. This allows to exemplify the expected response format. Additionally, we instructed the language models to produce their response in a particular format. Still, the tested models do not necessarily adhere to the instructions or the example format. Hence, to be able to extract the answer from the response, we developed a lenient *grammar* based parser, which would discard tokens if they did not fit expected token values. Figure 8 (Appendix) shows the grammar used for parsing the responses for our 8 tasks. For example, in the *progression task*, the response is a "progression_list", which is two lists, one for the positive effects and one for the negative effects. The response for the *justification* task is a "action_list", which is a list of actions (or the updated plan) and the response for the *validation* task is an "index" which is the index of the first inapplicable action. Using the grammar with a parser that discards tokens not consistent with the grammar helps significantly in post-processing these open-ended responses. The post-processed results are then evaluated according to the procedures described in Section 6. Our evaluation code is provided in the supplementary.

Focusing first on the small and medium size language models, Table 1 shows the accuracy of these models on the 8 tasks in our dataset. A conclusion we can draw from the figure is that there are three categories of tasks according to model performance. First,

| Model | app | areach | just | land | nexta | prog | reach | val |
|---|---|---|---|---|---|---|---|---|
| Granite 3.1 8B | 0.00 | 0.00 | 0.21 | 0.08 | 0.22 | 0.36 | **0.33** | 0.09 |
| Llama 3.1 8B | 0.00 | 0.00 | **0.22** | 0.06 | **0.25** | 0.40 | **0.33** | 0.13 |
| DeepSeek coder 33B | **0.02** | **0.02** | 0.21 | 0.10 | 0.17 | 0.42 | 0.18 | **0.15** |
| Granite 34B code | **0.02** | 0.00 | 0.17 | **0.11** | 0.18 | **0.43** | 0.28 | 0.12 |

Table 1: Small/medium size models performance, best results bolded.

*progression* seems to be the easiest tasks among the 8 tasks. Even so, the highest accuracy reached is 43% by Granite 34B code. The second category is the hard tasks of *applicability* and *action reachability*. In fact, none of these models could score above 2% for these two tasks. The third category includes the other 5 tasks with an average performance between 8.8% and 28.8%. Hence, ACPBench Hard seems to be a difficult dataset for the small and medium size language models.

Moving on to large size models, Table 2 shows the accuracy of the tested large language and reasoning models. Looking first at the language models, the top part of the table, we observe that there is no single model that outperforms all other models. GPT-4o is the top performer, with best results

in 5 out of 8 tasks, but it is outperformed by 5% by Mixtral 8x22B on the *reachability* task, and by DeepSeek V3 on the *action reachability* task by 4% and on the *justification* task by 11%. Among the large reasoning models (bottom part) the o1-preview model performs the best. Surprisingly, on the *justification* task the DeepSeek V3 language model remains the top performer. In all of our experiments we capped the maximum number of generated tokens at 1,000. For the GPT OSS models, however, we had to raise this limit to 4,000 because their reasoning chains were longer. Despite the higher token count and the resulting increase in inference cost, this did not translate into a performance boost. Similar to the small and medium size language models, there are tasks such as *progression* that is relatively easy for all models, and there are tasks such as *action reachability* that is difficult for all models, and even o1-preview achieves only 12%. For other tasks, some models do better than others. Our third observation is that with the exception of what seems to be the easiest for these models *progression* task, there are only three scores above 65%, and all of them are of reasoning models, indicating that the ACPBench Hard dataset is difficult even for large models.

Going deeper, we look into the performance of the best performing LLM, GPT-4o, across different domains. Figure 3 presents the performance of the top-performing language model GPT-4o on various tasks, across the existing domains. For other models performance we refer to the supplementary material. Observe that GPT-4o exhibits high performance on the *progression* task across most domains, showing somewhat lower accuracy in *satellite*, *alfworld*, and

| | Model | app | areach | just | land | nexta | prog | reach | val |
|---|---|---|---|---|---|---|---|---|---|
| LLMs | Mixtral 8x22B | 0.10 | 0.02 | 0.31 | 0.26 | 0.32 | 0.68 | 0.37 | 0.23 |
| | Llama 3.1 70B | 0.12 | 0.02 | 0.44 | 0.20 | 0.42 | 0.65 | 0.28 | 0.20 |
| | GPT-4o mini | 0.07 | 0.01 | 0.14 | 0.04 | 0.35 | 0.59 | 0.22 | 0.27 |
| | Llama 3.1 405B | 0.14 | 0.04 | 0.59 | 0.15 | 0.48 | 0.74 | 0.26 | 0.48 |
| | GPT-4o | 0.25 | 0.01 | 0.54 | 0.29 | 0.55 | 0.78 | 0.32 | 0.62 |
| | DeepSeek V3 | 0.21 | 0.05 | **0.65** | 0.12 | 0.47 | 0.76 | 0.32 | 0.56 |
| LRMs | o1 mini | 0.38 | 0.06 | 0.44 | 0.38 | 0.64 | 0.70 | 0.60 | **0.78** |
| | GPT OSS 20B | 0.03 | 0.09 | 0.14 | 0.47 | 0.62 | 0.72 | 0.50 | 0.14 |
| | GPT OSS 120B | 0.00 | **0.13** | 0.05 | 0.49 | 0.78 | 0.79 | **0.68** | 0.70 |
| | o1-preview | **0.44** | 0.12 | 0.46 | **0.56** | **0.80** | **0.89** | 0.66 | 0.26 |
| | DeepSeek R1 | 0.05 | 0.01 | 0.52 | 0.20 | 0.36 | 0.77 | 0.24 | 0.53 |

Table 2: Accuracy of Larger LLMs (top) and LRMs (bottom). **Bold** entries indicate the best overall model, while the underlined entries indicate the best among language (non-reasoning) models.

*floortile*. *Action reachability* and *landmarks* tasks, on the other hand, pose significant challenges to the model. The model consistently generated accurate answers for the *visitall* and *grid* domains across most tasks. In the remaining domains the performance varied across tasks, indicating that none of these domains are particularly easy for the models.

Finally, we evaluate the hardness of our proposed open-ended variants by comparing the performance of GPT-4o on these questions (gen) to the two formats from ACPBench: boolean (bool) and multiple choice (mcq). Figure 4 depicts the comparison in terms of model errors, the complement of the accuracy. As can be seen, model error in generative format of the task is significantly higher than bool and mcq except for the *validation* task. Note, we are showing results from one of the largest models for this analysis. The gap in performance is even more prolonged in other models.

## 7.2 OBSERVATIONS AND INSIGHTS

The **2-shots seemed to be sufficient** for the language models to mostly follow the instructions on the answer syntax. The number of parsing errors was typically negligible, with a single exception of Llama 3.1 8B on the applicable actions task, where in 4 cases the model made up objects that did not follow the naming convention in the question and in 5 cases the model ran out of context generating the same actions over and over again.

While atom and action reachability are very much related tasks, the latter seems to be much harder for the tested models. It is not surprising, as action reachability requires an additional reasoning step about the atoms in the action preconditions and their reachability. Further, while reachability focuses on single atoms, action reachability requires reasoning about the entire precondition, that often consists of multiple atoms. The reasoning should now account for their interplay, as they need to hold in the same reachable state. In fact, **action reachability seems to be consistently the most difficult task across the tested models**. Here, the models needed to provide an action that can never become applicable, in any reachable state or None if there are no such actions. Interestingly, only

the OpenAI models were able to correctly identify the latter case, and these were the majority of their correct answers. For instance, 9 out of 16 correct answers of o1-preview and all the 8 correct answers of o1 mini were *None*. Most of the other correct answers recognized that a block cannot be (un)stacked on itself, which accounts for 5 out of 6 correct answers for DeepSeek V3, 4 out of 5 for Llama 3.1 405B, and 4 out of 16 for o1-preview.

Surprisingly, **the second hardest task is action applicability**. While one of the core tasks in planning, it seems to be quite challenging even for the reasoning models, the largest of which scored 44%, while DeepSeek R1 scoring only 5%. The smaller language models fail on this task, with only DeepSeek coder 33B and Granite 34B code showing performance slightly above 0. Even the largest language models barely reach 25%. One reason for that is the strict requirement for producing precisely the set of all applicable actions. While missing actions can hinder completeness and cause missing existing solutions, extra made up actions can lead to producing incorrect solutions, which is arguably worse. Interestingly, if we only required not to make up actions, but allowed producing subsets of real answers, scoring according to Jaccard similarity, the score of the best performing model o1 preview would go up to 57%. The largest absolute increase would be for Mixtral, going from 10% to 38%. Using such action generation in a planner would correspond to loosing completeness, while keeping the soundness of the planner.

Looking at **the landmarks task**, among the smaller models, the best performing is Granite 34B. It gives quite uniform answers. Most of its correct answers are in the ferry domain, where it correctly identifies the ferry location as a landmark. In logistics, it recognizes a package need to be in the truck in the goal city before it can be delivered. The best performing large language model GPT-4o achieves 29% accuracy, producing a diverse set of answers. In ferry, it sometimes correctly identifies ferry being empty as a landmark, sometimes the need for a car to be onboard, and sometimes a ferry location was identified as a landmark. In logistics, trucks or packages at particular locations were identified as landmarks. In grid, it was more uniform, reporting in most cases holding a key as a landmark. In goldminer, all correctly identified landmarks were some locations being clear. The best performing reasoning model o1-preview achieves 56% accuracy. It correctly reports twice in satellite and 6 times in swap the absence of non-trivial landmarks. The model that is best at recognizing that is o1 mini, with 3 cases in visitall, 4 cases in satellite, 5 in swap, and once in alfworld. The only other model that can identify such cases is Mixtral 8x22B, with 3 cases in visitall.

The **justification task** is among the few tasks where the best among the language models is not GPT-4o, but DeepSeek V3, which performs better than even the reasoning models. The second best performer is Llama 3.1 405B. Interestingly, in multiple cases, Llama produces a valid plan, which is shorter than the given plan, but not its subsequence, and hence not a correct answer. In some cases, Llama produced a valid plan that changes the order of the given plan actions. Note, the aim of the justification task is not to produce a valid plan, but to recognize unnecessary actions on a sequence.

The **validation task** requires an index to be returned, which makes it harder to investigate the source of the mistakes made. Most language models do not perform well on this task, with only the largest models go above 30%. Interestingly, the reasoning model o1-preview scores much lower than o1 mini (26% vs 78%). To investigate the source of its mistakes, we check the absolute difference between the given answer and the correct one. In 86% of the cases that o1-preview incorrectly answered the question, it errored by 1. It is important to note that it was mistaken in both directions, giving both lower and higher than the correct indices. Note that several models favor the answers 1, 10, and 100. For the majority of smaller models, most their answers are one of these three numbers.

The **progression task** is mostly simple for the reasoning model o1-preview, which reaches 89%, but even this model makes mistakes. For example, it does not recognize reasonable positive effects, such as stacking a block on top of another block makes the top block clear. Not surprisingly, it misses less negative effects, such as taking a picture with a rover camera makes the camera not calibrated. Surprisingly, it invents unreasonable effects such as communicating the rock data from rover to the lander would empty the rover's store and delete the effect of rocks being analyzed at the waypoint.

## 7.3 REPRESENTATION

In all the experiments presented so far, our prompt included only the natural language representation of the domain and the problem as part of the context, "NL". Here, we experiment with two additional representations of the context: "PDDL", where the context included only the PDDL domain

and the PDDL problem, "PDDL+NL", where the context included both the natural language representation of the domain and problem as well as the PDDL representations. Table 3 presents the performance of DeepSeek V3. The results indicate that incorporating the PDDL domain alongside the natural language representation significantly improves performance. However, when the PDDL files are available, using a traditional planner may be more appropriate, rendering the use of an LLM unnecessary.

We look at the model performance on one of the computationally hardest tasks, the new *next action* task with the mixed PDDL+NL representation, where the model shows a higher than expected accuracy. To better understand the phenomena, Figure 5 shows the per-domain accuracy of DeepSeek V3 as a function of distance to goal. Note, the accuracy is

|          | app      | areach   | just     | land     | nexta    | prog     | reach    | val      | avg  |
|----------|----------|----------|----------|----------|----------|----------|----------|----------|------|
| NL       | 0.21     | 0.05     | 0.65     | 0.12     | 0.47     | 0.76     | 0.32     | 0.56     | 0.39 |
| PDDL     | 0.31     | 0.07     | **0.74** | **0.21** | 0.53     | 0.87     | 0.33     | 0.55     | 0.44 |
| PDDL+NL  | **0.32** | **0.09** | 0.68     | 0.19     | **0.60** | **0.88** | **0.37** | **0.61** | 0.47 |

Table 3: DeepSeek V3 comparison of 3 prompt representations: "NL", "PDDL", "PDDL+NL".

not uniform across domains. While, as expected, the accuracy decreases with the distance, we note some domains like ferry and satellite, where there is an increase for larger distances. The chances of randomly choosing a correct answer in these domains are low (See Table 7, Appendix).

# 8 CONCLUSIONS, LIMITATIONS, SOCIETAL IMPACT, AND FUTURE WORK

We have created ACP-Bench Hard, a benchmark of open-ended questions that reflect precisely the questions answered by symbolic planners during the planning process. We therefore believe that ACPBench Hard is a good benchmark for testing reasoning abilities that are required for planning. Based on an empirical investigation of a collection of large language models we conclude that even the largest and the best performing models have a very long way to go before they can reliably plan.

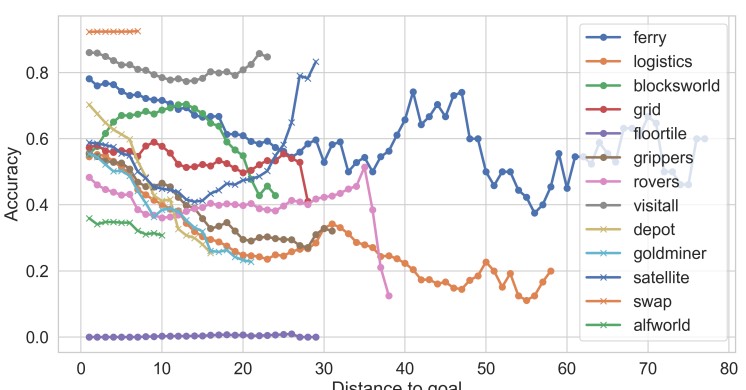

Figure 5: Domain-wise accuracy of DeepSeek V3 on next action, large evaluation set.

There are a few limitations to our work. First, the benchmark only captures a subset of planning problems. Second, our lenient evaluation only evaluates the first answer in the output. Third, we do not evaluate the reasoning process; only the final answers.

Improving the planning abilities of autonomous agents can have both positive and negative societal impact. Among the positive impact factors are increased efficiency and productivity of our society, which can also lead to negative factors such as job displacement and economic inequality. Being aware of these impact factors is the first step towards being able to exploit the positive impacts while mitigating the negative ones.

Creating training data for generative questions with a chain of thought, would be a promising area of future research, because for planning problems an instructive chain of thought is often not obvious. Another avenue for future research would be extending the benchmark to additional tasks such as object counting in a current state.

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

# 9 APPENDIX

# A TECHNICAL APPENDICES AND SUPPLEMENTARY MATERIAL

## A.1 RELATED WORK

Table 4 presents comparison of ACPBench Hard with other PDDL-based Planning Benchmarks. Existing benchmarks either focus on few tasks or few domains. For example, PlanBench has only 3 domains but 8 tasks, where as AutoPlanBench has 13 domains but only 1 task. Ac exception to this is Action Reasoning Bench which has reasonable number of domains and tasks, but they lack systematic evaluation of the generative questions. With addition of ACPBench Hard, now the ACPBench dataset contains all three formats of questions: generation, boolean and multi-choice questions. Most importantly, Table 4 presents the coverage of different tasks in the existing literature. As can be seen while the Action Applicability, Progression and Validation have some coverage in TRAC and ActionReasoningBench, the tasks like Reachability, Action Reachability, Justification, Landmark, and Next Actions are not really covered by existing benchmarks.

| Dataset | PlanBench | AutoPlanBench | TRAC | ARB | ACPBench | ACPBench Hard |
|---|---|---|---|---|---|---|
| # Tasks | 8 | 1 | 4 | 6 | 7 | 8 |
| # Domains | 3 (+variants) | 13 | 1 | 8 | 13 | 13 |
| NL templates | ✓ | ✗ | ✓ | ✓ | ✓ | ✓ |
| Evaluation | ✗ | ✗ | ↔ | ↔, LLM | ↔ | ✗ |
| Question Format | | | | | | |
| Generative | ✓ | ✓ | ✗ | ✓ | ✗ | ✓ |
| Boolean | ✗ | ✗ | ✓ | ✓ | ✓ | ✗ |
| MCQ | ✗ | ✗ | ✗ | ✗ | ✓ | ✗ |
| Tasks | | | | | | |
| Applicability | ✗ | ✗ | ✓ | ✓ | ✓ | ✓ |
| Progression | ✓ | ✗ | ✓ | ✓ | ✓ | ✓ |
| Reachability | ✗ | ✗ | ✗ | ✗ | ✓ | ✓ |
| Action Reachability | ✗ | ✗ | ✗ | ✗ | ✓ | ✓ |
| Validation | ✓ | ✗ | ✓ | ∼ | ✓ | ✓ |
| Justification | ✗ | ✗ | ✗ | ✗ | ✓ | ✓ |
| Landmark | ✗ | ✗ | ✗ | ✗ | ✓ | ✓ |
| Next Action | ✗ | ✗ | ✗ | ✗ | ✗ | ✓ |

Table 4: Comparison of ACPBench-hard with existing Planning Benchmarks. Evaluations are either using string matching ( ↔ ), symbolic tools ( ✗ ), or using another LLM ( LLM ).

**Action level**

**Applicability**

Identify all actions that can be
performed now, in a given state

*Failures: Hallucinated tools, blocked moves*

**Progression**

Predict everything that changes
after a particular action is taken.

*Failures: missing effects, Ignored side-effects*

**State level**

**Reachability**

What could possibly happen
and what is impossible from current state

*Fail: attempt impossible goals, invent states*

**Action Reachability**

Determine which actions could become
possible and which will never be

*Fail: invalid arguements to actions*

**Plan level**

**Validation**

Does the plan work?
Where does it break first?

*Fail: missed unmet pre-condition*

**Justification**

Simply the plan.
Remove redundancy

*Fail: Redundant plan, over-compression*

**Landmark**

Which milestones can
not be skipped?

*Fail: skip requisites, collapse dependencies*

**Next Action**

Pick a step that moves
you closer to the goal

*Fail: irrelevant drift*

Figure 6: Summary of 8 reasoning tasks included in ACPBench-Hard

## A.2 DATASET

Figure 6 summarizes the 8 tasks included in the ACPBench Dataset. Below, we provide one sample question for each of the task.

APPLICABILITY

Listing 1: Example of Applicability

```
{
    "id": -5674251047178000480,
    "group": "applicable_actions_gen",
    "context": "This is a ferry domain, where the task is to
        transport cars from their start to their goal locations,
        using a ferry. Each location is accessible by ferry from
        each other location. The cars can be debarked or boarded,
        and the ferry can carry only one car at a time. \nThere
        are 2 locations and 20 cars, numbered consecutively.
        \nCurrently, the ferry is at l0, with the car c2 on
        board. The cars are at locations as follows: c0, c3, c15,
        c10, c8, c6, and c9 are at l0; c18, c17, c16, c4, c19,
        c5, c13, c7, c1, c12, c14, and c11 are at l1. The
        available actions are: (sail ?from ?to) - travel by sea
        from location ?from to location ?to, (board ?car ?loc) -
        load the car ?car at location ?loc on to the ferry, and
        (debark ?car ?loc) - unload the car ?car from the ferry
        to location ?loc.",
    "question": "Generate the list of all ground actions that are
        applicable in this state.",
    "answer": [
      "(debark c2 l0)",
      "(sail l0 l1)"
    ],
    "PDDL_domain": "(define (domain ferry)\n    (:requirements :
        strips :typing)\n    (:types car location - object)\n
        (:predicates (at ?c - car ?l - location)  (at-ferry ?l -
        location)  (empty-ferry) (not-eq ?x ?y)  (on ?c - car))\n
        (:action board\n          :parameters (?car - car ?loc -
        location)\n         :precondition (and (at ?car ?loc)
        (at-ferry ?loc) (empty-ferry))\n          :effect (and (on
        ?car) (not (at ?car ?loc)) (not (empty-ferry)))\n    )\n
        (:action debark\n          :parameters (?car - car ?loc
        - location)\n         :precondition (and (on ?car)
        (at-ferry ?loc))\n          :effect (and (at ?car ?loc)
        (empty-ferry) (not (on ?car)))\n    )\n    (:action
        sail\n        :parameters (?from - location ?to -
        location)\n         :precondition (and (not-eq ?from ?to)
        (at-ferry ?from))\n         :effect (and (at-ferry ?to)
        (not (at-ferry ?from)))\n    )\n)",
    "PDDL_problem": "(define (problem ferry-l2-c20)\n    (:domain
        ferry)\n    (:requirements :strips :typing)\n
        (:objects c0 c1 c10 c11 c12 c13 c14 c15 c16 c17 c18 c19
        c2 c3 c4 c5 c6 c7 c8 c9 - car l0 l1 - location)\n
        (:init (at c0 l0) (at c1 l1) (at c10 l0) (at c11 l1) (at
        c12 l1) (at c13 l1) (at c14 l1) (at c15 l0) (at c16 l1)
        (at c17 l1) (at c18 l1) (at c19 l1) (at c3 l0) (at c4 l1)
        (at c5 l1) (at c6 l0) (at c7 l1) (at c8 l0) (at c9 l0)
        (at-ferry l0) (not-eq l0 l1) (not-eq l1 l0) (on c2))\n
        (:goal (and (at c0 l0) (at c1 l1) (at c2 l1) (at c3 l0)
        (at c4 l1) (at c5 l1) (at c6 l1) (at c7 l1) (at c8 l1)
```

```
        (at c9 l0) (at c10 l0) (at c11 l1) (at c12 l1) (at c13
        l0) (at c14 l1) (at c15 l0) (at c16 l1) (at c17 l1) (at
        c18 l1) (at c19 l1)))\n)"
}
```

PROGRESSION

Listing 2: Example of Progression

```
{
  "id": 297440160406485545,
  "group": "progression_gen",
  "context": "This is a ferry domain, where the task is to
      transport cars from their start to their goal locations,
      using a ferry. Each location is accessible by ferry from
      each other location. The cars can be debarked or boarded,
      and the ferry can carry only one car at a time. \nThere
      are 2 locations and 10 cars, numbered consecutively.
      \nCurrently, the ferry is at l1, with the car c2 on
      board. The cars are at locations as follows: c5, c9, and
      c4 are at l1; c3, c0, c1, c6, c7, and c8 are at l0. The
      available propositions are: (at-ferry ?l) - The ferry is
      at ?l location, (at ?c ?l) - Car ?c is at location ?l,
      (empty-ferry) - The ferry is empty, and (on ?c) - Ferry
      has car ?c on board.",
  "question": "Break down the outcomes of performing the action
      \"debark car c2 to location l1 from the ferry\" into two
      lists, positive effects and negative effects. Positive
      effects are the propositions that are false in the
      current state but will become true after performing the
      action. Negative effects are the propositions that are
      true in the current state and will become false after
      performing the action.",
  "answer": {
    "pos": [
      "(empty-ferry)",
      "(at c2 l1)"
    ],
    "neg": [
      "(on c2)"
    ]
  },
  "PDDL_domain": "(define (domain ferry)\n    (:requirements :
      strips :typing)\n    (:types car location - object)\n
      (:predicates (at ?c - car ?l - location)  (at-ferry ?l -
      location)  (empty-ferry) (not-eq ?x ?y)  (on ?c - car))\n
          (:action board\n           :parameters (?car - car ?loc -
      location)\n         :precondition (and (at ?car ?loc)
      (at-ferry ?loc) (empty-ferry))\n          :effect (and (on
      ?car) (not (at ?car ?loc)) (not (empty-ferry)))\n     )\n
          (:action debark\n          :parameters (?car - car ?loc
      - location)\n        :precondition (and (on ?car)
      (at-ferry ?loc))\n         :effect (and (at ?car ?loc)
      (empty-ferry) (not (on ?car)))\n     )\n       (:action
      sail\n        :parameters (?from - location ?to -
      location)\n        :precondition (and (not-eq ?from ?to)
      (at-ferry ?from))\n         :effect (and (at-ferry ?to)
      (not (at-ferry ?from)))\n     )\n)",
```

```
    "PDDL_problem": "(define (problem ferry-l2-c10)\n    (:domain
        ferry)\n    (:requirements :strips :typing)\n
        (:objects c0 c1 c2 c3 c4 c5 c6 c7 c8 c9 - car l0 l1 -
        location)\n    (:init (at c0 l0) (at c1 l0) (at c3 l0)
        (at c4 l1) (at c5 l1) (at c6 l0) (at c7 l0) (at c8 l0)
        (at c9 l1) (at-ferry l1) (not-eq l0 l1) (not-eq l1 l0)
        (on c2))\n    (:goal (and (at c0 l0) (at c1 l0) (at c2
        l1) (at c3 l1) (at c4 l0) (at c5 l0) (at c6 l0) (at c7
        l0) (at c8 l0) (at c9 l1)))\n)"
}
```

REACHABILITY

Listing 3: Example of Reachability

```
{
    "id": 6900855040701022305,
    "group": "reachable_atom_gen",
    "context": "This is a ferry domain, where the task is to
        transport cars from their start to their goal locations,
        using a ferry. Each location is accessible by ferry from
        each other location. The cars can be debarked or boarded,
        and the ferry can carry only one car at a time. \nThere
        are 2 locations and 5 cars, numbered consecutively.
        \nCurrently, the ferry is at l0, with the car c2 on
        board. The cars are at locations as follows: c3, c0, and
        c1 are at l0; c4 is at l1. The available propositions
        are: (at-ferry ?l) - The ferry is at ?l location, (at ?c
        ?l) - Car ?c is at location ?l, (empty-ferry) - The ferry
        is empty, and (on ?c) - Ferry has car ?c on board.",
    "question": "What proposition can never hold in any
        potentially reachable state?",
    "answer": [],
    "PDDL_domain": "(define (domain ferry)\n    (:requirements :
        strips :typing)\n    (:types car location - object)\n
        (:predicates (at ?c - car ?l - location)  (at-ferry ?l -
        location)  (empty-ferry) (not-eq ?x ?y)  (on ?c - car))\n
        (:action board\n        :parameters (?car - car ?loc -
        location)\n        :precondition (and (at ?car ?loc)
        (at-ferry ?loc) (empty-ferry))\n        :effect (and (on
        ?car) (not (at ?car ?loc)) (not (empty-ferry)))\n    )\n
        (:action debark\n        :parameters (?car - car ?loc
        - location)\n        :precondition (and (on ?car)
        (at-ferry ?loc))\n        :effect (and (at ?car ?loc)
        (empty-ferry) (not (on ?car)))\n    )\n    (:action
        sail\n        :parameters (?from - location ?to -
        location)\n        :precondition (and (not-eq ?from ?to)
        (at-ferry ?from))\n        :effect (and (at-ferry ?to)
        (not (at-ferry ?from)))\n    )\n)",
    "PDDL_problem": "(define (problem ferry-l2-c5)\n    (:domain
        ferry)\n    (:requirements :strips :typing)\n
        (:objects c0 c1 c2 c3 c4 - car l0 l1 - location)\n
        (:init (at c0 l0) (at c1 l0) (at c3 l0) (at c4 l1)
        (at-ferry l0) (not-eq l0 l1) (not-eq l1 l0) (on c2))\n
        (:goal (and (at c0 l1) (at c1 l0) (at c2 l1) (at c3 l0)
        (at c4 l0)))\n)"
}
```

ACTION REACHABILITY

Listing 4: Example of Action Reachability

```
{
    "id": -255316435894028208,
    "group": "reachable_action_gen",
    "context": "This is a ferry domain, where the task is to
        transport cars from their start to their goal locations,
        using a ferry. Each location is accessible by ferry from
        each other location. The cars can be debarked or boarded,
        and the ferry can carry only one car at a time. \nThere
        are 2 locations and 20 cars, numbered consecutively.
        \nCurrently, the ferry is at l0, with the car c2 on
        board. The cars are at locations as follows: c12, c18,
        c5, c4, c14, c19, c16, c11, c1, and c7 are at l1; c13,
        c6, c10, c17, c9, c3, c8, c0, and c15 are at l0. The
        available actions are: (sail ?from ?to) - travel by sea
        from location ?from to location ?to, (board ?car ?loc) -
        board the car ?car at the location ?loc, and (debark ?car
        ?loc) - debark the car ?car to location ?loc from the
        ferry.",
    "question": "What action can never become applicable, in any
        state reachable from the current state?",
    "answer": [
      "(sail l0 l0)"
    ],
    "PDDL_domain": "(define (domain ferry)\n    (:requirements :
        strips :typing)\n    (:types car location - object)\n
        (:predicates (at ?c - car ?l - location)  (at-ferry ?l -
        location)  (empty-ferry) (not-eq ?x ?y)  (on ?c - car))\n
        (:action board\n        :parameters (?car - car ?loc -
        location)\n        :precondition (and (at ?car ?loc)
        (at-ferry ?loc) (empty-ferry))\n        :effect (and (on
        ?car) (not (at ?car ?loc)) (not (empty-ferry)))\n    )\n
        (:action debark\n        :parameters (?car - car ?loc
        - location)\n        :precondition (and (on ?car)
        (at-ferry ?loc))\n        :effect (and (at ?car ?loc)
        (empty-ferry) (not (on ?car)))\n    )\n    (:action
        sail\n        :parameters (?from - location ?to -
        location)\n        :precondition (and (not-eq ?from ?to)
        (at-ferry ?from))\n        :effect (and (at-ferry ?to)
        (not (at-ferry ?from)))\n    )\n)",
    "PDDL_problem": "(define (problem ferry-l2-c20)\n    (:domain
        ferry)\n    (:requirements :strips :typing)\n
        (:objects c0 c1 c10 c11 c12 c13 c14 c15 c16 c17 c18 c19
        c2 c3 c4 c5 c6 c7 c8 c9 - car l0 l1 - location)\n
        (:init (at c0 l0) (at c1 l1) (at c10 l0) (at c11 l1) (at
        c12 l1) (at c13 l0) (at c14 l1) (at c15 l0) (at c16 l1)
        (at c17 l0) (at c18 l1) (at c19 l1) (at c3 l0) (at c4 l1)
        (at c5 l1) (at c6 l0) (at c7 l1) (at c8 l0) (at c9 l0)
        (at-ferry l0) (not-eq l0 l1) (not-eq l1 l0) (on c2))\n
        (:goal (and (at c0 l0) (at c1 l1) (at c2 l1) (at c3 l0)
        (at c4 l1) (at c5 l1) (at c6 l1) (at c7 l1) (at c8 l1)
        (at c9 l0) (at c10 l0) (at c11 l1) (at c12 l1) (at c13
        l0) (at c14 l1) (at c15 l0) (at c16 l1) (at c17 l1) (at
        c18 l1) (at c19 l1)))\n)"
}
```

VALIDATION

Listing 5: Example of Validation

```
{
    "id": 4896696031890359153,
    "group": "validation_gen",
    "context": "This is a ferry domain, where the task is to
        transport cars from their start to their goal locations,
        using a ferry. Each location is accessible by ferry from
        each other location. The cars can be debarked or boarded,
        and the ferry can carry only one car at a time. \nThere
        are 2 locations and 5 cars, numbered consecutively.
        \nCurrently, the ferry is at l0 location and it is empty.
        The cars are at locations as follows: c3, c1, c0, and c2
        are at l0; c4 is at l1. The goal is to reach a state
        where the following facts hold: Car c3 is at location l0,
        Car c4 is at location l0, Car c1 is at location l0, Car
        c0 is at location l1, and Car c2 is at location l1. The
        available actions are: (sail ?from ?to) - sail from
        location ?from to location ?to, (board ?car ?loc) - board
        the car ?car at the location ?loc, and (debark ?car ?loc)
        - debark the car ?car from the ferry to location ?loc.",
    "question": "What is the first inapplicable action in the
        next sequence of actions: \"(board c2 l0) (debark c2 l0)
        (board c2 l0) (sail l0 l1) (board c2 l1) (board c4 l1)
        (sail l1 l0) (debark c4 l0) (board c0 l0) (sail l0 l1)
        (debark c0 l1) (sail l1 l0)\"?",
    "answer": 4,
    "PDDL_domain": "(define (domain ferry)\n    (:requirements :
        strips :typing)\n    (:types car location - object)\n
        (:predicates (at ?c - car ?l - location)  (at-ferry ?l -
        location)  (empty-ferry) (not-eq ?x ?y)  (on ?c - car))\n
            (:action board\n            :parameters (?car - car ?loc -
        location)\n         :precondition (and (at ?car ?loc)
        (at-ferry ?loc) (empty-ferry))\n          :effect (and (on
        ?car) (not (at ?car ?loc)) (not (empty-ferry)))\n    )\n
            (:action debark\n           :parameters (?car - car ?loc
        - location)\n         :precondition (and (on ?car)
        (at-ferry ?loc))\n         :effect (and (at ?car ?loc)
        (empty-ferry) (not (on ?car)))\n    )\n    (:action
        sail\n        :parameters (?from - location ?to -
        location)\n         :precondition (and (not-eq ?from ?to)
        (at-ferry ?from))\n          :effect (and (at-ferry ?to)
        (not (at-ferry ?from)))\n    )\n)",
    "PDDL_problem": "(define (problem ferry-l2-c5)\n    (:domain
        ferry)\n    (:requirements :strips :typing)\n
        (:objects c0 c1 c2 c3 c4 - car l0 l1 - location)\n
        (:init (at c0 l0) (at c1 l0) (at c2 l0) (at c3 l0) (at c4
        l1) (at-ferry l0) (empty-ferry) (not-eq l0 l1) (not-eq l1
        l0))\n    (:goal (and (at c0 l1) (at c1 l0) (at c2 l1)
        (at c3 l0) (at c4 l0)))\n)"
}
```

JUSTIFICATION

Listing 6: Example of Justification

```
{
```

```
"id": -1219355986766168268,
"group": "action_justification_gen",
"context": "This is a ferry domain, where the task is to
    transport cars from their start to their goal locations,
    using a ferry. Each location is accessible by ferry from
    each other location. The cars can be debarked or boarded,
    and the ferry can carry only one car at a time. \nThere
    are 2 locations and 2 cars, numbered consecutively.
    \nCurrently, the ferry is at l0 location and it is empty.
    The cars are at locations as follows: c1 and c0 are at
    l0. The available actions are: (sail ?from ?to) - sail
    from location ?from to location ?to, (board ?car ?loc) -
    board the car ?car at the location ?loc, and (debark ?car
    ?loc) - debark the car ?car to location ?loc from the
    ferry. The goal is to reach a state where the following
    facts hold: Car c1 is at location l1 and Car c0 is at
    location l1.",
"question": "Simplify the plan \"(board c1 l0) (sail l0 l1)
    (sail l1 l0) (sail l0 l1) (debark c1 l1) (sail l1 l0)
    (sail l0 l1) (sail l1 l0) (board c0 l0) (sail l0 l1)
    (debark c0 l1) (board c1 l1) (debark c1 l1)\" by removing
    either a single action or a pair of consecutive actions,
    while still maintaining a valid plan. Provide the
    resulting simplified plan.",
"answer": [
  [
    "(sail l0 l1)",
    "(sail l1 l0)",
    "-1"
  ],
  [
    "(sail l1 l0)",
    "(sail l0 l1)",
    "-1"
  ],
  [
    "(sail l1 l0)",
    "(sail l0 l1)",
    "-1"
  ],
  [
    "(sail l0 l1)",
    "(sail l1 l0)",
    "-1"
  ],
  [
    "(board c1 l1)",
    "(debark c1 l1)",
    "-1"
  ]
],
"PDDL_domain": "(define (domain ferry)\n    (:requirements :
    strips :typing)\n    (:types car location - object)\n
    (:predicates (at ?c - car ?l - location)  (at-ferry ?l -
    location)  (empty-ferry) (not-eq ?x ?y)  (on ?c - car))\n
        (:action board\n        :parameters (?car - car ?loc -
    location)\n        :precondition (and (at ?car ?loc)
    (at-ferry ?loc) (empty-ferry))\n        :effect (and (on
    ?car) (not (at ?car ?loc)) (not (empty-ferry)))\n    )\n
```

```
        (:action debark\n        :parameters (?car - car ?loc
    - location)\n        :precondition (and (on ?car)
    (at-ferry ?loc))\n        :effect (and (at ?car ?loc)
    (empty-ferry) (not (on ?car)))\n    )\n    (:action
    sail\n        :parameters (?from - location ?to -
    location)\n        :precondition (and (not-eq ?from ?to)
    (at-ferry ?from))\n        :effect (and (at-ferry ?to)
    (not (at-ferry ?from)))\n    )\n)",
    "PDDL_problem": "(define (problem ferry-l2-c2)\n    (:domain
    ferry)\n    (:requirements :strips :typing)\n
    (:objects c0 c1 - car l0 l1 - location)\n    (:init (at
    c0 l0) (at c1 l0) (at-ferry l0) (empty-ferry) (not-eq l0
    l1) (not-eq l1 l0))\n    (:goal (and (at c0 l1) (at c1
    l1)))\n)"
}
```

LANDMARK

Listing 7: Example of Landmark

```
{
    "id": 5196274110229243987,
    "group": "landmarks_gen",
    "context": "This is a ferry domain, where the task is to
        transport cars from their start to their goal locations,
        using a ferry. Each location is accessible by ferry from
        each other location. The cars can be debarked or boarded,
        and the ferry can carry only one car at a time. \nThere
        are 2 locations and 10 cars, numbered consecutively.
        \nCurrently, the ferry is at l1, with the car c6 on
        board. The cars are at locations as follows: c9, c4, and
        c2 are at l1; c1, c7, c3, c0, c5, and c8 are at l0. The
        goal is to reach a state where the following facts hold:
        Car c9 is at location l1, Car c4 is at location l0, Car
        c6 is at location l0, Car c1 is at location l0, Car c7 is
        at location l0, Car c2 is at location l1, Car c3 is at
        location l1, Car c0 is at location l0, Car c5 is at
        location l0, and Car c8 is at location l0. The available
        propositions are: (at-ferry ?l) - The ferry is at ?l
        location, (at ?c ?l) - Car ?c is at location ?l,
        (empty-ferry) - The ferry is empty, and (on ?c) - Car ?c
        is on board the ferry.",
    "question": "Generate a non-trivial fact landmark, one that
        does not hold in the initial state or goal.",
    "answer": {
      "yes": [
        "(on c3)",
        "(on c4)",
        "(at-ferry l0)",
        "(empty-ferry)"
      ],
      "no": [
        "(at c2 l0)",
        "(at c5 l1)",
        "(at c1 l1)",
        "(at c8 l1)",
        "(at c6 l1)",
        "(on c1)",
        "(at c0 l1)",
```

```
            "(on c5)",
            "(on c9)",
            "(on c2)",
            "(on c7)",
            "(at c7 l1)",
            "(on c0)",
            "(on c8)",
            "(at c9 l0)"
        ]
    },
    "PDDL_domain": "(define (domain ferry)\n    (:requirements :
        strips :typing)\n    (:types car location - object)\n
        (:predicates (at ?c - car ?l - location)  (at-ferry ?l -
        location)  (empty-ferry) (not-eq ?x ?y)  (on ?c - car))\n
           (:action board\n          :parameters (?car - car ?loc -
        location)\n        :precondition (and (at ?car ?loc)
        (at-ferry ?loc) (empty-ferry))\n         :effect (and (on
        ?car) (not (at ?car ?loc)) (not (empty-ferry)))\n    )\n
           (:action debark\n          :parameters (?car - car ?loc
        - location)\n        :precondition (and (on ?car)
        (at-ferry ?loc))\n          :effect (and (at ?car ?loc)
        (empty-ferry) (not (on ?car)))\n    )\n      (:action
        sail\n       :parameters (?from - location ?to -
        location)\n         :precondition (and (not-eq ?from ?to)
        (at-ferry ?from))\n        :effect (and (at-ferry ?to)
        (not (at-ferry ?from)))\n    )\n)",
    "PDDL_problem": "(define (problem ferry-l2-c10)\n    (:domain
        ferry)\n    (:requirements :strips :typing)\n
        (:objects c0 c1 c2 c3 c4 c5 c6 c7 c8 c9 - car l0 l1 -
        location)\n    (:init (at c0 l0) (at c1 l0) (at c2 l1)
        (at c3 l0) (at c4 l1) (at c5 l0) (at c7 l0) (at c8 l0)
        (at c9 l1) (at-ferry l1) (not-eq l0 l1) (not-eq l1 l0)
        (on c6))\n    (:goal (and (at c0 l0) (at c1 l0) (at c2
        l1) (at c3 l1) (at c4 l0) (at c5 l0) (at c6 l0) (at c7
        l0) (at c8 l0) (at c9 l1)))\n)"
}
```

NEXT ACTION

Listing 8: Example of Next Action

```
{
    "id": -6003545580663971528,
    "group": "goal_closer_gen",
    "context": "This is a ferry domain, where the task is to
        transport cars from their start to their goal locations,
        using a ferry. Each location is accessible by ferry from
        each other location. The cars can be debarked or boarded,
        and the ferry can carry only one car at a time. \nThere
        are 2 locations and 5 cars, numbered consecutively.
        \nCurrently, the ferry is at l1 location and it is empty.
        The cars are at locations as follows: c0, c4, and c1 are
        at l0; c3 and c2 are at l1. The goal is to reach a state
        where the following facts hold: Car c0 is at location l1,
        Car c4 is at location l0, Car c1 is at location l0, Car
        c3 is at location l0, and Car c2 is at location l1. The
        available actions are: (sail ?from ?to) - travel by sea
        from location ?from to location ?to, (board ?car ?loc) -
        board the car ?car at the location ?loc, and (debark ?car
```

```
        ?loc) - debark the car ?car from the ferry to location
        ?loc.",
    "question": "What is the next action that takes us towards
        the goal?",
    "answer": {
      "yes": [
        "(board c3 l1)"
      ],
      "no": [
        "(board c2 l1)",
        "(sail l1 l0)"
      ],
      "maybe": [],
      "opt": "6"
    },
    "PDDL_domain": "(define (domain ferry)\n    (:requirements :
        strips :typing)\n    (:types car location - object)\n
        (:predicates (at ?c - car ?l - location)  (at-ferry ?l -
        location)  (empty-ferry) (not-eq ?x ?y)  (on ?c - car))\n
          (:action board\n           :parameters (?car - car ?loc -
        location)\n         :precondition (and (at ?car ?loc)
        (at-ferry ?loc) (empty-ferry))\n           :effect (and (on
        ?car) (not (at ?car ?loc)) (not (empty-ferry)))\n     )\n
          (:action debark\n           :parameters (?car - car ?loc
        - location)\n         :precondition (and (on ?car)
        (at-ferry ?loc))\n           :effect (and (at ?car ?loc)
        (empty-ferry) (not (on ?car)))\n     )\n     (:action
        sail\n         :parameters (?from - location ?to -
        location)\n         :precondition (and (not-eq ?from ?to)
        (at-ferry ?from))\n         :effect (and (at-ferry ?to)
        (not (at-ferry ?from)))\n     )\n)",
    "PDDL_problem": "(define (problem ferry-l2-c5)\n    (:domain
        ferry)\n    (:requirements :strips :typing)\n
        (:objects c0 c1 c2 c3 c4 - car l0 l1 - location)\n
        (:init (at c0 l0) (at c1 l0) (at c2 l1) (at c3 l1) (at c4
        l0) (at-ferry l1) (empty-ferry) (not-eq l0 l1) (not-eq l1
        l0))\n    (:goal (and (at c0 l1) (at c1 l0) (at c2 l1)
        (at c3 l0) (at c4 l0)))\n)"
}
```

## A.3 GENERATION PROCESS

Figure 7 illustrates the overall generation process of the ACPBench-Hard dataset.

### APPLICABILITY

While constructing the action applicability dataset, we impose a bound on the number of applicable actions in a state. We only keep the states when $|A(s)|$ is $\leq 100$. While the bound is high, in most cases the number of applicable actions are $\leq 10$. Table 5 presents the Max and Mean number of Applicable Actions.

### EVALUATION

Table 6 show the complexity of evaluating an answer per task.

## A.4 GRAMMAR FOR PARSING THE MODEL RESPONSE

Figure 8 shows the grammar used for parsing the responses for our 8 tasks.

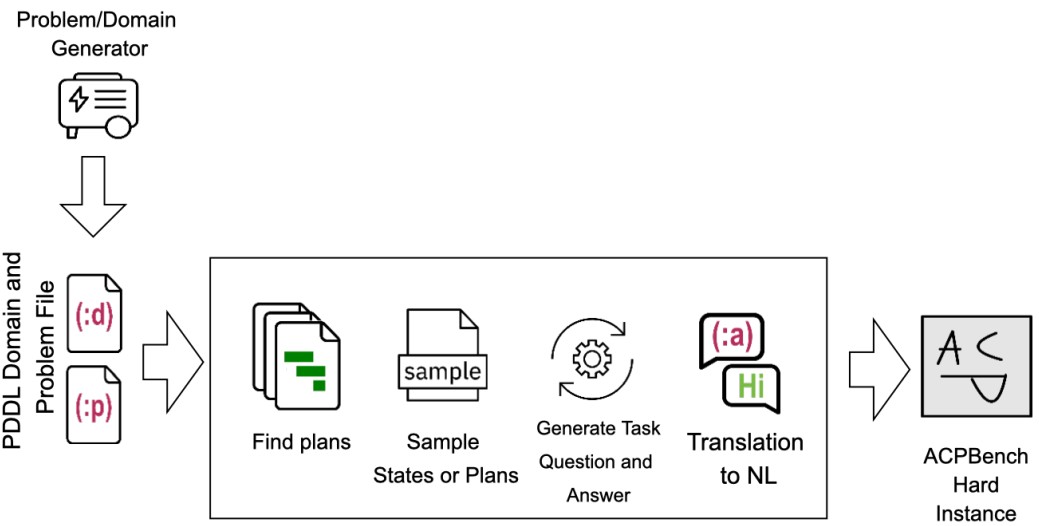

Figure 7: High-level Workflow of Generating ACPBench Hard Dataset. **1.** We use the Problem/-Domain Generator when available to generate PDDL Problem and Domain Files. For Alfworld domain, we skip this step and directly use the PDDL Domain and Problem file made available in the Alfworld repo. **2.** Given a PDDL Problem and Domain File, we use a top-K planner (Katz et al., 2020) to find multiple plans. **3** Depending on the task, we either sample plans from the obtained plans or sample intermediate states on these plans. **4.** We generate question and answer for the task symbolically. **5.** Translate the Symbolic task to Natural Language using Templates. **6.** Save the Natural Language question and the information needed to evaluate the answer to the ACPBench Hard Dataset

| Domain | # Applicable Actions | |
| --- | --- | --- |
| | **Max** | **Mean** |
| ferry | 10 | 5.00 |
| logistics | 10 | 9.50 |
| blocksworld | 9 | 4.60 |
| grid | 6 | 4.00 |
| floortile | 10 | 9.50 |
| grippers | 9 | 6.10 |
| rovers | 10 | 7.90 |
| visitall | 4 | 2.80 |
| depot | 10 | 9.70 |
| goldminer | 4 | 2.80 |
| satellite | 8 | 8.00 |
| swap | 49 | 37.80 |
| alfworld | 99 | 66.20 |

Table 5: Number of correct applicable actions in the Applicability task dataset.

| Task | Complexity |
|------|-----------|
| Applicability (App) | $O(|A|)$ |
| Progression (Prog) | $O(|F|)$ |
| Reachability (Reach) | PSPACE-complete |
| Action Reachability (AReach) | PSPACE-complete |
| Validation (Val) | $O(1)$ |
| Justification (Just) | $O(|\pi||F|)$ |
| Landmarks (Land) | PSPACE-complete |
| Next Action (NextA) | PSPACE-complete |

Table 6: Computational complexity of answer evaluation per task.

```
NAME:/[a-zA-Z][a-zA-Z0-9-_]*/
LPAR:"("
RPAR:")"
WS:/[ \n]/
LSPAR:"["
RSPAR:"]"
COMMA:","
action_none: "None"
action_name: LPAR NAME (WS NAME)* RPAR
act: action_name | action_none
action_list: (action_name WS?)*
prog_list: action_name* (COMMA action_name)*
progression_list: LSPAR prog_list RSPAR LSPAR prog_list RSPAR
index:/[0-9]+[0-9]*/
```

Figure 8: Grammar used for parsing the model response.

## A.5 EXPERIMENTAL SETUP

All our experiments were carried our using temperature 0 for replicability. We leverages LM Evaluation Harness (Gao et al., 2024) framework for our experiments and use default configurations for the experiments. The YAML config file for our experiments is provided in the supplemental material. We highlight the most important hyperparameters below.

We only focused on 2-shot prompting in this work for two reasons: 1. LRMs used in our experiments generate Chain-of-Thought (COT), so specialized prompt is not required for that. The previous work (see Fig. 5 of (Kokel et al., 2025)) did not show much advantage of COT prompts, only in-context examples showed real advantage. We did not include hybrid approaches because the primary goal of this work is to propose a challenging benchmark for natural language tasks for Language Models.

```
num_fewshot: 2
generation_kwargs:
  until:
    - "\n\n\n\n"
    - "\n\n"
    - "**Question**:"
    - "**Question:**"
    - "Q:"
  do_sample: false
  max_gen_toks: 1000
  temperature: 0.0
```

## A.6 NEXT ACTION, SPLIT BY DOMAIN

One of the tasks, specifically the new *next action* task shows a higher than expected accuracy for such a computationally hard task. To better understand the phenomena, Table 7 shows the per-domain split. The bolded and underlined values depict the top accuracy values per domain across the tested models. Note, the accuracy is not uniform across domains. Some more complex domains such as *depot* shows better performance across models than easy domains such as *grippers*, which is hard for all models. There are some domains, such as *logistics* and *ferry* that are hard for some models

| Model | depot | goldminer | satellite | swap | alfworld | ferry | logistics | blocks | grid | floortile | grippers | rovers | visitall |
|---|---|---|---|---|---|---|---|---|---|---|---|---|---|
| Mixtral 8x22B | 0.4 | 0.2 | 0.4 | 0.8 | 0.1 | 0.3 | 0.4 | 0.3 | 0.3 | 0.1 | 0.2 | 0.4 | 0.3 |
| Llama 3.1 70B | 0.6 | 0.4 | 0.5 | 0.7 | 0.1 | 0.2 | 0.4 | 0.5 | 0.5 | 0.4 | **0.3** | 0.3 | 0.5 |
| GPT-4o mini | 0.4 | 0.4 | 0.4 | 0.7 | 0.0 | 0.5 | 0.1 | 0.4 | 0.6 | 0.1 | 0.2 | 0.3 | 0.5 |
| DeepSeek V3 | 0.5 | 0.4 | 0.2 | **1.0** | 0.2 | 0.7 | **0.8** | 0.6 | 0.6 | 0.0 | 0.1 | 0.5 | 0.5 |
| Llama 3.1 405B | 0.6 | 0.4 | 0.4 | **1.0** | 0.4 | 0.4 | 0.6 | 0.2 | 0.6 | 0.0 | **0.3** | 0.6 | 0.7 |
| GPT-4o | 0.6 | 0.7 | 0.5 | 0.9 | 0.4 | 0.7 | 0.6 | 0.6 | 0.7 | 0.1 | 0.2 | 0.8 | 0.4 |
| DeepSeek R1 | 0.4 | 0.5 | 0.2 | 0.9 | 0.0 | 0.4 | 0.6 | 0.5 | 0.4 | 0.0 | 0.1 | 0.2 | 0.5 |
| o1 mini | **0.9** | 0.8 | 0.5 | 0.9 | 0.5 | **1.0** | 0.4 | 0.6 | **0.9** | 0.3 | 0.2 | 0.4 | **0.9** |
| o1-preview | 0.8 | **0.9** | **0.8** | 0.9 | **0.9** | **1.0** | 0.8 | **0.8** | **0.9** | **0.8** | 0.1 | **0.9** | 0.8 |
| random (full set) | 0.11 | 0.43 | 0.04 | 0.21 | 0.03 | 0.19 | 0.14 | 0.39 | 0.36 | 0.22 | 0.20 | 0.26 | 0.49 |

Table 7: Accuracy of large size models on the *next action* task, split by domain. Bold entries indicate the best overall model, while the underlined entries indicate the best language model. Last row indicates the ratio of the actions that are correct answers among the applicable actions, the performance of a random selector among applicable actions.

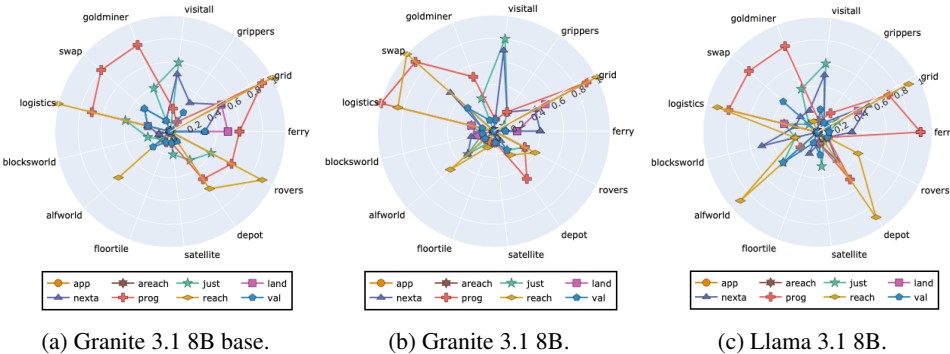

(a) Granite 3.1 8B base.  (b) Granite 3.1 8B.  (c) Llama 3.1 8B.

Figure 9: Domain-wise accuracy of small size language models.

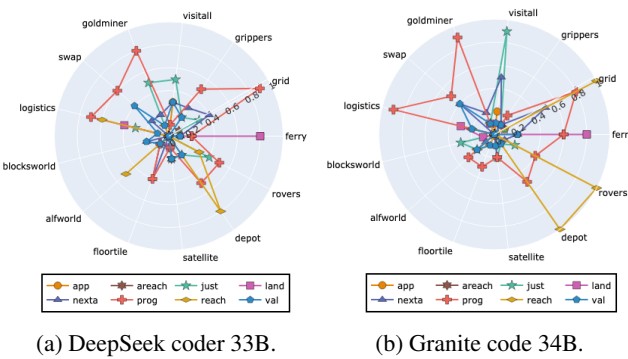

(a) DeepSeek coder 33B.  (b) Granite code 34B.

Figure 10: Domain-wise accuracy of small and medium size language models.

and easy for other. There are domains like *floortile* and *alfworld* that are very hard for almost all models.

## A.7   DOMAIN-WISE ACCURACY OF TESTED MODELS

Figures 9 and 10 show the domain-wise accuracy of tested small and medium language models, while Figure 11 presents the domain-wise accuracy of tested large language and reasoning models.

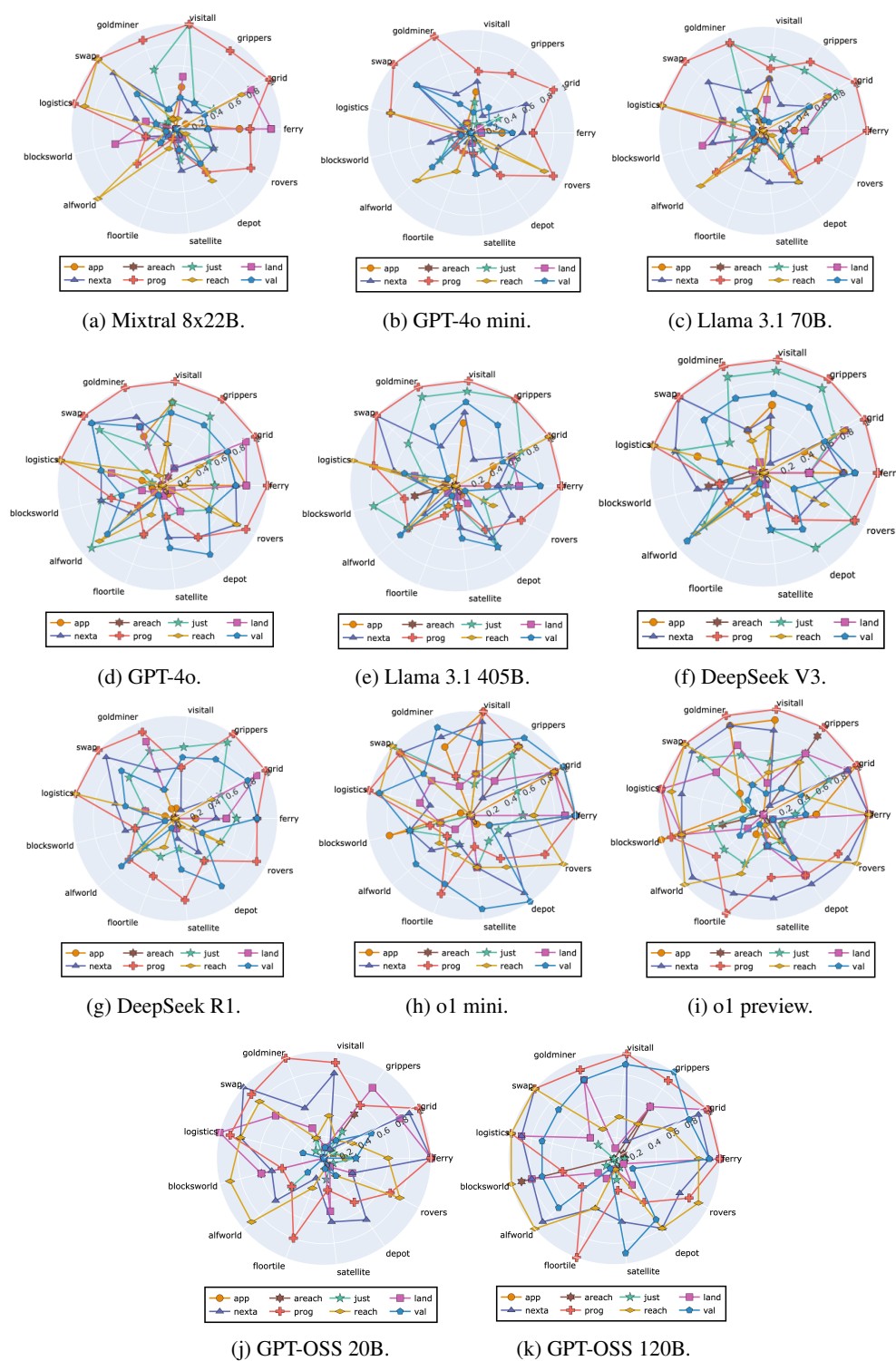

Figure 11: Domain-wise accuracy of large language and reasoning models.

## A.8 LESS STRICT EVALUATION

Our evaluation of applicability and progression tasks expect complete correct answers. This is because, partially correct answers in the progression task result in an incorrect resulting state, which is

detrimental to reliable planning. Partially correct answers are somewhat possible only in the action applicability task. If the answer includes incorrect actions, it can result in unsound solutions produced. However, if the answer is incomplete, the result is a loss of completeness, which makes it still possible to produce a sound solution. In such cases, we can have a less strict evaluation metric.

So in addition to the exact set match metric, a less-strict metric like Jaccard similarity score can also be leveraged. Table 8 provide domain-wise comparison of both metrics for the best-performing model `o1-preview`.

| Domain | Exact Match | Jaccard Score |
|---|---|---|
| alfworld | 0 | 0.20 |
| blocksworld | 1 | 1 |
| depot | 0.3 | 0.54 |
| ferry | 0.5 | 0.5 |
| floortile | 0.2 | 0.73 |
| goldminer | 0.9 | 0.9 |
| grid | 1 | 1 |
| grippers | 0 | 0 |
| logistics | 0.2 | 0.2 |
| rovers | 0.2 | 0.57 |
| satellite | 0.2 | 0.29 |
| swap | 0.3 | 0.63 |
| visitall | 0.9 | 0.9 |
| Mean | 0.44 | 0.57 |

Table 8: Comparison of two exact-match metric with a less strict metric, Jaccard Similarity, for O1-Preview on Applicability task.

## A.9 LIMITATIONS

Our work aims to propose a reliable benchmark the evaluates LLM's ability to reason about action, change and planning. However, there are a few limitations to our work.

- Captures a subset of planning problems: classical domains - closed world, deterministic dynamics, full observability, described in a standard planning language PDDL.
- Requires specifying the corresponding natural language mapping of facts and actions per domain.
- Generation and validation are both computationally challenging and therefore time consuming. The validation makes use of lenient grammar, resulting in possibly capturing a wrong part of the answer, when multiple answers are given in one response.
- Evaluating generated output is hard, while we counted on the model to not produce multiple conflicting answers, LLMs are known to do so.
- The evaluations only measures the final answer and do not evaluate the reasoning process of LLMs.
- We use same set of prompts to evaluate all the LLM. Having different prompts and approaches might result in different performance on the benchmark.

