# OpenReview forum: "ACPBench Hard: Unrestrained Reasoning about Action, Change, and Planning"
_ICLR.cc/2026/Conference — ICLR 2026 Poster_

### Official Review · Reviewer_Zfe9 · 2025-10-28

**Soundness:** 4
**Presentation:** 3
**Contribution:** 2
**Rating:** 4
**Confidence:** 5

**Summary:**

The paper describes and novel dataset on planning tasks which are supposed to be harder. The paper describes the creation and details the included task. It contains 13 planning task with each 8 examples if I read the paper correctly. They describe that none of the datasets can be solved by the best models and reach about up to 60% . The task are in natural language translated from pddl.

**Strengths:**

1. New and harder planning dataset.
2. Detailed evaluation on a larger set of models for small to large LLMs.
3. Detailed analysis of results.

**Weaknesses:**

The datasets is seems too small per task to obtain good confidence intervals. I would prefer per task much more examples, via the automatic process they describe, you could produce per planning problems thoustands of samples automatically. I would strongly suggetest to provide more. These seem expensive to run but per tokens prices fall and for small models they run the costs a very small. You could define subsets.

The task should be provide as both NL and PDDL. The models understand PDDL well. NL is not anymore needed. The translation is per template and needs backtranslation for testing.

Valmeekam et al. used thinking models and solved many of these task. Blocksworld with 20+ blocks can be now solved by these models. Even Valmeekam used not the latest models of the thinking models now a year ago at Neurisp and the newer once might solve all the provided problems, I guess. The authors could have tried these with GPT-5 or Gemini 2.5 thinking models. These experiments seem missing.

**Questions:**

1. Could you provide per planning problem 1000 samples? Or is this too much effort in case can you explain why? I think this would be ideal.
2. It would be nice to try with the latest thinking models using NL and PDDL and evaluate both. Would strengthen the claim that these models can not solve those for some time.
3. Could you provide the statistics about the models in a table, e.g. how much planning steps in average, how distributed, how many samples, etc.

---

> ### Author Response · Authors · 2025-11-14
>
> **W1**
>
> Valmeekam et al. deal with with the plan generation problem, on a limited number of domains, mostly the blocksworld domain. Indeed, the newest language models have been trained extensively on these domains to generate plans, but it does not mean that they are able to solve the tasks we pose in this work or handle domains other than blocksworld and logistics. Our experiments with frontier models demonstrate this clearly. While exhaustive experimentation across all models is infeasible, the subset we evaluate is a well-chosen and representative one. Once the paper is accepted, other researchers are of course free to experiment with additional models, though we suspect the results will be similar. We conjecture that pure LLM-based methods may not be sufficient to achieve consistently strong performance on these tasks.
>
>
> **Q1**
>
> There seems to be a misunderstanding regarding the size of the data set we experiment on. Our evaluation is performed on 130 questions for each of the 8 tasks (total of 1040 questions).
>
> We first generated a set of thousands of examples per domain (out of 13 domains) and per task (out of 8 tasks). We then chose randomly 10 examples per domain (130 in total) for each of the 8 tasks. We have now updated the paper (beginning of section 6.1) to include this info.
>
> We can automatically generate any number of tasks, and a large number of examples per task.  We do provide both the natural language and the PDDL representation.
>
> **Q2**
>
> Please see section 6.3 for the comparison of different representations, NL only, PDDL only, and NL+PDDL, of one of the largest available open models DeepSeek V3, showing that the model can take advantage of the structural representation, as well as of the natural text. Note that we would like to get a good performance on these tasks with as small and efficient models as possible, since in real planning scenarios these tasks need to be performed millions of times during the planning process for every planning task.
>
> **Q3**
>
> To clarify, these are atomic reasoning tasks that need to be performed by a planner to be able to plan reliably. In other words, each task is a single step task.

---

### Official Review · Reviewer_wXMk · 2025-10-31

**Soundness:** 3
**Presentation:** 4
**Contribution:** 3
**Rating:** 8
**Confidence:** 5

**Summary:**

This paper introduces ACPBench Hard, a benchmark designed to evaluate LLMs on open-ended questions about planning. It builds upon the prior ACPBench work which only included boolean and multiple-choice questions, with the difference being that this benchmark requires models to generate complete answers that automated planners would produce during the planning process. Specifically, the benchmark includes eight reasoning tasks across thirteen planning domains: applicability, progression, reachability, action reachability, validation, justification, landmarks, and next action. The authors develop symbolic validators for each task and evaluate fifteen models ranging from 8B to 405B parameters, including reasoning models like o1. Their central finding is that even frontier language models perform poorly, with most scoring below 65% on most tasks, indicating substantial gaps in planning reasoning capabilities.

**Strengths:**

The paper makes an important advance in evaluating the planning/reasoning abilities of LLMs and LRMs by shifting from discriminative to generative evaluation, which more faithfully represents the actual decision-making processes that (symbolic) planners perform. This transition is non-trivial because generative tasks require models to produce answers from large action spaces rather than selecting from pre-curated options, making the benchmark more realistic for assessing practical planning capabilities.

Using symbolic validators is also a substantial strength of the benchmark. Unlike other approaches that use other LLMs for evaluation (which introduces noise and potential biases), these validators provide deterministic and theoretically grounded correctness verification. The authors thoughtfully discuss computational complexity for both the tasks themselves and their validation procedures, with several tasks being PSPACE-complete.

The empirical evaluation is comprehensive, testing fifteen models including small (8B), medium (33-34B), large (70B-405B), and reasoning models (o1, DeepSeek R1), providing good coverage of the current model landscape. The inclusion of recent reasoning models is particularly valuable for understanding whether extended inference computation translates to improved planning reasoning.

The qualitative analysis also provides insights into specific failure modes, such as models recognizing blocks cannot be stacked on themselves but struggling with more subtle applicability constraints, or o1-preview's systematic off-by-one errors in the validation task. These concrete observations are more informative than aggregate statistics alone.

The investigation of representation formats, i.e., combinations of NL and PDDL descriptions, is a good additional experiment. Maybe not surprisingly, the PDDL+NL representation substantially improves performance, which may have some implications for how planning problems should be presented to LLMs.

**Weaknesses:**

One of the immediate weaknesses I notice in the paper is on the evaluation side. Namely:

- The main paper lacks details about exact prompts used, model hyperparameters (temperature, top-p,), random seeds, and how many independent runs were conducted. The paper states a maximum of 1,000 tokens was used (4,000 for GPT-OSS models) but does not specify whether models used greedy decoding or sampling with what parameters. Given the stochastic nature of LLMs, these details are essential for reproducibility.

- The paper employs only 2-shot prompting without exploring alternative prompting strategies. Given the complexity of these tasks, this seems limiting. The paper does not investigate whether chain-of-thought prompting, task decomposition, or additional examples might improve performance. The reasoning models presumably use some form of extended reasoning internally, but the regular LLMs were not given explicit instructions to reason step-by-step in the prompts. At least, that's what can be inferred from the main paper. Additionally, the "lenient grammar-based parser" that discards non-conforming tokens may reject valid answers expressed in unexpected formats. The paper acknowledges evaluating only the first answer when models might produce multiple, but does not quantify how often parsing failures occur versus genuine incorrect answers.

- The representation experiment reveals that PDDL+NL substantially improves performance (at least for DeepSeek V3). But why does providing PDDL help? Is it because the structured representation reduces ambiguity? Do models pattern-match on PDDL syntax? Does this suggest LLMs are better at parsing structured representations than natural language descriptions? This result might have implications for prompt engineering and practical deployment, meaning that a deeper analysis including ablations and attention visualization will be beneficial (however, one can argue this is beyond the scope of this paper).

**Questions:**

1. Concerning natural language generation, how exactly were PDDL problems translated to natural language? I understand you're extending the previous ACPBench, but was this process automated using templates, manually created, or some combination?

2. What hyperparameter values, i.e., temperature and top-p, were used in the experiments? How sensitive are results to different hyperparameter values? Were multiple independent runs conducted to assess variance?

3. Did you experiment with alternative prompting strategies such as explicit CoT reasoning, task decomposition, or providing worked examples for multi-step reasoning?

4. Can you quantify how often the grammar-based parser failed to extract answers versus models providing incorrect answers? Did you manually inspect a sample of parsing failures to ensure the parser is not rejecting valid answers?

5. Can you provide more detailed analysis of why providing PDDL helps? Does it reduce ambiguity, provide structure that models can pattern-match, or serve a different function?

---

> ### Author Response · Authors · 2025-11-14
>
> **Q1**
>
> The translation of PDDL problem to natural language is done using NL Template. We indicate that in Table 4.  We have updated the paper and also highlight it in the Fig. 6 caption.
>
> **Q2**
>
> All our experiments were conducted with temp 0 and default hyper-parameters with LM-Eval Harness to ensure replicability. We provide the config files in the supplemental material. We will also include it in the appendix. Given time and resource constraints, we prioritized evaluating a broader set of models rather than performing multiple runs with varying seeds, temperature, and top-p values on a single model.
>
> **Q3**
>
> Thank you, we only focused on 2-shot prompting in this work for two reasons:
> * Most reasoning models essentially generate COT, so specialized prompt is not required for that.
> * The previous work (see Fig. 5 of Kokel et. al., AAAI 2025) did not show much advantage of COT prompts, only in-context examples showed real advantage.
>
> **Q4**
>
> Yes, we manually inspected the answers for a few models, especially for models that had a score of 0 for a particular task. We found that the 2-shot examples were sufficient for  most language models to follow the required answer format. Parsing errors were negligible, with one exception: Llama 3.1 8B on the applicable actions task (9 out of 130 questions). In this case, the model occasionally invented objects that did not match the naming convention (4 instances) and, in 5 cases, ran out of context, repeatedly generating the same actions.
>
> **Q5**
>
> We hypothesize that the models were trained on a significant amount of PDDL data that includes instances and plans, which helped the models generating correctly structured actions and facts.

---

> ### Comment · Reviewer_wXMk · 2025-11-26
> **Response to the Authors' comments**
>
> The authors' response addressed my questions satisfactorily, and I am willing to keep my score as it is.

---

### Official Review · Reviewer_UptW · 2025-10-31

**Soundness:** 3
**Presentation:** 4
**Contribution:** 3
**Rating:** 8
**Confidence:** 5

**Summary:**

This paper introduces ACPBench Hard, a benchmark specifically designed for evaluating the Reasoning about Actions and Change (RAC) capabilities of Large Language Models (LLMs). Compared to existing RAC benchmarks (such as TRAC, ActionReasoningBench, and ACPBench), ACPBench Hard features a richer set of tasks and transforms reasoning tasks into generative questions. Furthermore, the authors designed dedicated symbolic validation mechanisms for each tasks to avoid the uncertainties associated with using LLMs for evaluation. Then, the authors conducted a comprehensive evaluation of 15 LLMs and reasoning models of varying scales using this benchmark. The experimental results clearly demonstrate that even SOTA models perform poorly on these fundamental RAC tasks, indicating that current LLMs are not yet capable of being reliably used as components in symbolic planners or as standalone planners.

**Strengths:**

1. Compared to other RAC datasets, the ACPBench Hard dataset features more challenging tasks and provides rigorous formal definitions for each task.

2. The use of symbolic solvers for answer evaluation avoids potential information loss that might occur when relying solely on LLMs for assessment.

3. The paper goes beyond evaluating end-to-end planning capabilities and delves into underlying, atomic-level reasoning. The authors accurately analyze and pinpoint specific reasons for LLM failures in these tasks (for example, LLMs cannot even reliably list all executable actions in a given state).

4. The evaluation is comprehensive, covering models of different sizes, both LLMs and LRMs, as well as open-source and proprietary models. The domains used in the benchmark are also extensive.

**Weaknesses:**

1. The evaluation employed a 2-shot prompting strategy. It remains unclear whether this prompting approach is optimal for eliciting planning and reasoning capabilities in models. More sophisticated strategies (such as ToT and GoT, SC, etc) might yield different outcomes. Besides, it is not clear whether using examples from the target domain would help.

2. The generative vs. mcq comparison is limited. Figure 4 only compares the GPT-4o model, making it uncertain to determine whether the increased difficulty is a universal phenomenon or specific to certain models.

**Questions:**

Q1: There is an inconsistency in task names. Line 381 mentions "atom reachability" - does this refer to Task 3 "reachability"?

Q2: Using symbolic solvers to verify generative answers usually requires a formalization process. I am curious whether using only a grammar parser is sufficient to accomplish this translation, given the diversity of LLM responses. Do parsing failures occur during evaluation? If so, have you considered adopting LLM-based evaluation as a backup strategy when parsing fails?

Q3: During evaluation, did you set specific temperature parameters for the LLMs, or were default values used throughout?

Q4: NextA is an interesting task. How exactly did you compute the heuristic value h*(s) for states?

---

> ### Author Response · Authors · 2025-11-14
>
> **Q1**
>
> Thank you for carefully reviewing our work. Yes “atom reachability” in line 381 refers to the reachability task. We have made it consistent now.
>
> **Q2**
>
> We believe LLM-based evaluation brings its own set of challenges. As mentioned in line 376, the 2-shot examples were generally sufficient for language models to follow the required answer format. Parsing errors were negligible, with one exception: Llama 3.1 8B on the applicable actions task. In this case, the model occasionally invented objects that did not match the naming convention (4 instances) and, in 5 cases, ran out of context, repeatedly generating the same actions. So we did not see the need to delve into LLM-based evaluation.
>
> **Q3**
>
> For all the evaluations, we use temperature 0. We have now added details about hyper parameters in the Appendix Sec A.5
>
> **Q4**
>
> We use a cost-optimal planner to find a plan $\pi$ for the planning task with the state $s$ being the initial state and use its cost $cost(\pi)$, which is exactly the value of the perfect heuristic $h^{\ast}(s)$.

---

### Official Review · Reviewer_2qTa · 2025-11-01

**Soundness:** 3
**Presentation:** 2
**Contribution:** 2
**Rating:** 4
**Confidence:** 3

**Summary:**

The paper introduces ACPBench Hard, a benchmark dataset for evaluating LLMs ability to perform generative reasoning tasks related to automated planning. Building upon the existing ACPBench, the authors create open-ended, generative versions of 7 existing tasks plus one new task. The benchmark spans 13 PDDL planning domains and includes symbolic validators for each task. The authors conduct an extensive empirical evaluation of 15 LLMs and reasoning models, revealing that even state-of-the-art models perform poorly on most tasks, with many scores below 65%.

**Strengths:**

* The paper addresses a significant problem by requiring generative responses rather than classification, which better reflects real-world planner requirements.
* The paper tests 15 models across different scales and types, providing empirical coverage
* The description of different tasks in the paper is clear and structured. The discussion of the computational complexity for each task is thorough.

**Weaknesses:**

* The evaluation only considers the first answer provided by the model. This might underestimate a model's capabilities if it generates a correct answer later in its output.
* The model needs to produce all actions’ effects correctly to be scored. This evaluation is strict. Partial credits may be considered as other evaluation metric.
* The baselines of the evaluation are not comprehensive enough. Only 2-shot prompting is included. No CoT prompting or hybrid approaches with solvers are included.
* The paper is text heavy. More illustrations can be added to aid understanding. The high level workflow looks simple and clear but is not in the main texts.

**Questions:**

* What would results look like under less strict metrics?
* What is the dataset size per task? For each task, can you include more information such as some data examples, example input/output, failure cases?

---

> ### Author Response · Authors · 2025-11-14
>
> **W1**
>
> A model might generate the correct answer later on. When the model produces multiple responses, identifying the intended response is non-trivial and is a research question in its own right. Our current approach to parse LLM output uses a grammar-based parser that only accepts the first answer that follows the grammar. This is in line with common practice, such as using Pydantic to parse LLM outputs into Python objects.
>
> **W2**
>
> Partially correct answers in the progression task result in an incorrect resulting state, which is detrimental to reliable planning. Partially correct answers are somewhat possible only in the action applicability task. If the answer includes incorrect actions, it can result in unsound solutions produced. However, if the answer is incomplete, the result is a loss of completeness, which makes it still possible to produce a sound solution. In such cases, we can have a less strict evaluation metric. We discuss this case in lines 400-403.
>
> **W3**
>
> We only focused on 2-shot prompting in this work for two reasons:
> * LRMs used in our experiments generate COT, so specialized prompt is not required for that.
> * The previous work (see Fig. 5 of Kokel et. al., AAAI 2025) did not show much advantage of COT prompts, only in-context examples showed real advantage.
>
> We did not include hybrid approaches because the primary goal of this work is to propose a challenging benchmark for natural language tasks for Language Models.

---

> > ### Author Response · Authors · 2025-11-14
> >
> > **Q1**
> >
> > A less-strict metric only makes sense for the Action applicability task. We discuss using the Jaccard Score for action applicability in lines 400-403. We provide domain-wise comparison of both metrics in the table below for the best-performing model o1-preview.
> >
> > | Metric | alfworld | blocksworld | depot | ferry | floortile | goldminer | grid | grippers | logistics | rovers | satellite | swap | visitall | Mean |
> > |:--------------|-----------:|--------------:|---------:|--------:|------------:|------------:|-------:|-----------:|------------:|---------:|------------:|---------:|-----------:|-----------:|
> > | Exact Match | 0 | 1 | 0.3 | 0.5 | 0.2 | 0.9 | 1 | 0 | 0.2 | 0.2 | 0.2 | 0.3 | 0.9 | 0.44 |
> > | Jaccard Score | 0.20 | 1 | 0.54 | 0.5 | 0.73 | 0.9 | 1 | 0 | 0.2 | 0.57 | 0.29 | 0.63 | 0.9 | 0.57 |
> >
> > **Q2**
> >
> > ACPBench Hard dataset consists of 130 questions for each task (10 questions per domain), total of 1040 questions. We have now updated the paper (beginning of section 6.1) to include this info. Below we provide 1 example for each task. Due to space constraints we did not include it in the paper. We have added the examples to the Appendix.

---

### Meta-Review · Area_Chair_JdLt · 2026-01-06

**Summary:**

This is a benchmark paper for evaluating the planning capabilities of LLMs/LRMs. The main points of criticism that were given by all reviewers concerned the experimental evaluation. In terms of breadth and also the statistical methods used.

**Reviewer Concerns:**

The authors did not fully address all the concerns by the reviewers. However, they clarified some of their experimental choices and "promised" to add further experimental evaluations. If accepted I would encourage the authors to come through on their promises and include some of the explanations/details that were asked for by the reviewers in the paper.

**Reviewer Scores:**

I do not think any of the reviewers would have changed their scores (3/4 had confidence scores of 5).

---

### Decision · Program_Chairs · 2026-01-26

Accept (Poster)